# From Schwinger to Wightman:
# all conformal 3-point functions
# in momentum space

Marc Gillioz

*SISSA, via Bonomea 265, 34136 Trieste, Italy*

**Abstract**

All conformal correlation functions of 3 scalar primary operators are constructed in an axiomatic way, relying only on conformal symmetry and causality. The construction makes use of the R-product and its analyticity properties in Minkowski momentum space. The R-product is completely determined by a system of partial differential equations (the Ward identities for conformal transformations) and a boundary condition (permutation symmetry), up to an OPE coefficient. Wightman functions and time-ordered products are then derived from the R-product using operator identities, and the Schwinger function is recovered after a Wick rotation to Euclidean space. This construction does not rely on the Fourier transform of position-space correlation functions.

# 1 Introduction

In conformal field theory (CFT), all correlation functions are determined by data characterizing the 2- and 3-point functions only, as higher-point function can be reduced using the operator product expansion (OPE). Moreover, 2- and 3-point functions are fixed by conformal symmetry up to a finite number of parameters. For scalar primary operators this is just the scaling dimensions of the operators involved, a choice of normalization for the 2-point function, and a single real number determining the strength of the 3-point interaction.

This is true both in the position-space representation, where most of the 2- and 3-point functions have been explicitly constructed [1–7], and in the momentum-space representation, related to the former by Fourier transform. It is also true both in Euclidean space and in Minkowski space-time. In this work, we show how the Minkowski momentum-space 2- and 3-point correlation functions can be constructed from symmetry arguments only, without referring to the known position-space representation. The construction relies solely on a few ingredients that are essentially equivalent to the Wightman axioms of quantum field theory [8], plus the constraints imposed by conformal symmetry.[1] These

---

[1]All of our assumptions can be traced back to the axioms of the Euclidean conformal bootstrap in position space [9, 10], but we want to emphasize here that it is not necessary to go through an analytic continuation from Euclidean to Minkowski space followed by a Fourier transform to obtain useful results.

ingredients are:

(1) Conformal symmetry, in the form of Ward identities for correlation functions of primary operators. This includes Poincaré and scale symmetry, but also special conformal symmetry expressed as a system of second-order partial differential equations whose general solution is a linear combination of Appell $F_4$ functions.

(2) Analyticity of the R-product (a generalization of the concept of retarded commutator involving more than 2 operators) in the forward tube. This property is a direct consequence of the micro-causality condition in Minkowski space-time.[2]

(3) Permutation symmetry of the R-products, specifically eqs. (3.2) and (3.3) given below. Note that this second equation involves Wightman functions.

(4) The spectral condition for Wightman functions, which is equivalent to the requirement that only states of positive energy are part of the Hilbert space.

These axioms are sufficient to construct all 2- and 3-point R-products unequivocally, up to the normalization factor and OPE coefficient mentioned already. They also lead to similar results for Wightman functions. Finally, time-ordered products are obtained using a fifth ingredient:

(5) The identities (2.27) and (5.14) relating the T- and R-products. These can be established as identities satisfied by position-space correlation functions; alternatively they can be viewed as a definition of the T-product valid in momentum space.

The correlators obtained in this way are found to match precisely with known results obtained taking the Fourier transform of position-space correlators [11, 12]. Moreover, the analytic structure of the time-ordered correlator suggests a simple Wick rotation that reproduces the known Schwinger function in Euclidean momentum space [13, 14]. However, the results for many orderings of the operators are new. They apply in all generality in any space-time dimension $d \geq 2$. Note that we do not discuss situations in which all momenta are simultaneously complex: in doing so, we mostly avoid subtleties associated with functions of multiple complex variables. Nevertheless, the analyticity properties of the R-product taken one momentum at a time are essential to perform continuations between real kinematic domains that are otherwise disconnected.

Finally, table 1 at the end of this work contains an exhaustive list of all correlators constructed with our method. The reader who is not interested in the details of the derivation but rather concerned with using a specific correlator is invited to consult it.

## 1.1 Notation

Since multi-point correlation functions involve several variables that have themselves more than one component, it is useful to set up the stage and introduce some notation that simplifies the presentation of our results.

---

[2]It does not strictly follow from the Wightman axioms since R-product defined in position space is not a tempered distribution, and its Fourier transform can be ill-defined in some cases. However, we shall see that our momentum-space definition leads to an unequivocal R-product, and that exceptions can be treated by analytic continuation in the scaling dimensions of the operators.

First of all, because of momentum conservation (or equivalently translation symmetry), correlators involving $n$ primary operators are functions of $n - 1$ independent momenta: they are always proportional to a delta function imposing momentum conservation, and we will therefore make use of the "double bracket" notation

$$\langle 0| \phi_1(k_1) \cdots \phi_n(k_n) |0\rangle = (2\pi)^d \delta^d(k_1 + \ldots + k_n) \langle\!\langle \phi_1(k_1) \cdots \phi_n(k_n) \rangle\!\rangle. \tag{1.1}$$

One could substitute $k_n = -k_1 - \ldots - k_{n-1}$ in the correlator on the right-hand side to emphasize that $\langle\!\langle \phi_1(k_1) \cdots \phi_n(k_n) \rangle\!\rangle$ only depends on the momenta $k_1$ to $k_{n-1}$, but it is often more convenient to make use of $n$ momenta with the implicit assumption that they sum up to zero: for instance, in a 3-point function we might use $k_1$, $k_2$ and $k_3$ and write the three Lorentz invariants compactly as $k_1^2$, $k_2^2$ and $k_3^2$, instead of using the explicit form $k_3^2 = (k_1 + k_2)^2$.

Even though we do not make explicit use of the Fourier transform, it is worth specifying our definition of momentum-space operators in terms of the more familiar position-space operators (decorated with a tilde to avoid confusion):

$$\phi(k) = \int d^d x \, e^{ik \cdot x} \widetilde{\phi}(x). \tag{1.2}$$

Correlators in the momentum-space representation are therefore the Fourier transform of correlators in the position-space representation,

$$\langle 0| \phi_1(k_1) \cdots \phi_n(k_n) |0\rangle = \int d^d x_1 \cdots d^d x_n \, e^{i(k_1 \cdot x_1 + \ldots + k_n \cdot x_n)} \langle 0| \widetilde{\phi}_1(x_1) \cdots \widetilde{\phi}_n(x_n) |0\rangle. \tag{1.3}$$

Using invariance of the correlation function under translations, this gives an alternative definition of the "double bracket" as the Fourier transform of a position-space correlator in which one of the points is held fixed at the origin of the coordinate system,

$$\langle\!\langle \phi_1(k_1) \cdots \phi_n(k_n) \rangle\!\rangle = \int d^d x_1 \cdots d^d x_{n-1} \, e^{i(k_1 \cdot x_1 + \ldots + k_{n-1} \cdot x_{n-1})}$$
$$\times \langle 0| \widetilde{\phi}_1(x_1) \cdots \widetilde{\phi}_{n-1}(x_{n-1}) \widetilde{\phi}_n(0) |0\rangle. \tag{1.4}$$

This definition will prove useful when discussing conformal Ward identities, as primary operators inserted at the origin have simple transformation properties.

Note that we use the "mostly minus" metric $k^2 = (k^0)^2 - (k^1)^2 - \ldots - (k^{d-1})^2$, and denote the spatial part of a vector in bold font, i.e. $k^2 = (k^0)^2 - \mathbf{k}^2$. A space-like vector has therefore $k^2 < 0$. In fact, to avoid having to specify whether a vector is space-like or time-like, we use the following conventions:

- $p_i$ indicates a time-like vector inside the forward light cone, i.e. $p_i^2 > 0$ with positive energy $p_i^0 > |\mathbf{p}_i|$. Vectors inside the backward light cone are simply denoted $-p_i$.

- $q_i^\mu$ indicates a space-like vector, i.e. $q_i^2 < 0$.

- $k_i^\mu$ indicates a vector that can either be space-like or time-like, or even complex.

For instance, the correlator $\langle\!\langle \phi_1(-p_1)\phi_2(q_2)\phi_3(p_3) \rangle\!\rangle$ denotes a Wightman function in which the spectral condition is satisfied: the operator $\phi_3$ carries a momentum $p_3$ that lies inside the forward light cone, and so does the product of operators $\phi_2 \times \phi_3$, as their total momentum $q_2 + p_3 = p_1$ (by momentum conservation) is again inside the forward light cone.

# 2 Warming up with the 2-point function

We begin with a discussion of 2-point functions of scalar primary operators. This provides a simple framework in which all the key ideas used later with the 3-point function appear.

## 2.1 The retarded commutator

The retarded commutator, or R-product of two scalar primary operators, is the correlation function denoted

$$\langle\!\langle R[\phi_1(k), \phi_2(-k)]\rangle\!\rangle. \tag{2.1}$$

It transform covariantly under conformal symmetry: it is Lorentz-invariant, and obeys Ward identities associated to scale transformations,

$$\left(-k^\mu \frac{\partial}{\partial k^\mu} + \Delta_1 + \Delta_2 - d\right) \langle\!\langle R[\phi_1(k), \phi_2(-k)]\rangle\!\rangle = 0, \tag{2.2}$$

as well as special conformal transformations

$$\left(-2k^\nu \frac{\partial^2}{\partial k_\mu \partial k^\nu} + k^\mu \frac{\partial^2}{\partial k_\nu \partial k^\nu} + 2(\Delta_1 - d)\frac{\partial}{\partial k_\mu}\right) \langle\!\langle R[\phi_1(k), \phi_2(-k)]\rangle\!\rangle = 0. \tag{2.3}$$

Here $\Delta_1$ and $\Delta_2$ are the scaling dimensions of the operators $\phi_1$ and $\phi_2$ respectively. Moreover, the R-product has the property of being analytic in the forward tube

$$k \in \mathbb{T}_+ \qquad \Leftrightarrow \qquad \text{im } k^0 > |\text{im } \mathbf{k}|. \tag{2.4}$$

These properties are motivated by the definition of the retarded commutator in position space,

$$\langle 0| \, \text{R}[\widetilde{\phi}_1(x_1), \widetilde{\phi}_2(x_2)] \, |0\rangle = \theta(x_1^0 - x_2^0) \, \langle 0| \, [\widetilde{\phi}_1(x_1), \widetilde{\phi}_2(x_2)] \, |0\rangle + \text{contact terms}, \tag{2.5}$$

where $\theta$ is the Heaviside step function, i.e. $\theta(a) = 1$ if $a > 0$ and zero otherwise. This R-product vanishes whenever the separation between $x_1$ and $x_2$ is space-like by micro-causality; it also vanishes when $x_1$ is in the past of $x_2$, and thus

$$\langle 0| \, \text{R}[\widetilde{\phi}_1(x), \widetilde{\phi}_2(0)] \, |0\rangle = 0 \qquad \forall \, x^0 < |\mathbf{x}|. \tag{2.6}$$

The contact terms mentioned in eq. (2.5) only have support at $x_1 = x_2$: they are proportional to $\delta^d(x_1 - x_2)$ or derivatives thereof, and must be adjusted to make the Fourier transform of this correlator exist. Following the convention (1.4), this means that we identify

$$\langle\!\langle R[\phi_1(k), \phi_2(-k)]\rangle\!\rangle = \int d^d x \, e^{ik\cdot x} \, \langle 0| \, \text{R}[\widetilde{\phi}_1(x), \widetilde{\phi}_2(0)] \, |0\rangle. \tag{2.7}$$

Since the integrand in this expression only has support in the future light cone $x^0 \geq |\mathbf{x}|$, and since it decays exponentially fast at large $x^0$ provided that $\text{im } k^0 \geq |\text{im } \mathbf{k}|$, this integral defines an analytic function of $k$ in the forward tube (2.4). Similarly, the Ward identities (2.2) and (2.3) can be derived from the transformation properties of the operators in

position space. For instance, special conformal transformations are generated by the operator $K^\mu$ satisfying

$$[K^\mu, \widetilde{\phi}_i(x)] = \left( 2x^\mu x^\nu \frac{\partial}{\partial x^\nu} - x^2 \frac{\partial}{\partial x_\mu} + 2\Delta_i x^\mu \right) \widetilde{\phi}_i(x). \tag{2.8}$$

Since the vacuum state is by assumption conformally invariant, and since primary operators inserted at the origin do not transform under special conformal transformations, $[K^\mu, \widetilde{\phi}(0)] = 0$, one can deduce that the 2-point correlation function must satisfy the Ward identity[3]

$$\left( 2x^\mu x^\nu \frac{\partial}{\partial x^\nu} - x^2 \frac{\partial}{\partial x_\mu} + 2\Delta_1 x^\mu \right) \langle 0| \, \mathrm{R}[\widetilde{\phi}_1(x), \widetilde{\phi}_2(0)] \, |0\rangle = 0. \tag{2.9}$$

This can be turned into a second-order differential equation for the Fourier integral (2.7), yielding precisely the Ward identity (2.3). The Ward identities associated with Lorentz and scale symmetries are derived in the same way, and translation symmetry is already built in the definition (2.7), as we chose to place the operator $\phi_2$ at the origin of the coordinate system.

The 2-point function of scalar primary operators is well known in CFT, and the Fourier transform (2.7) could easily be computed directly. The only subtlety has to do with fixing the contact terms, but even this is simple as it can be bypassed using analytic continuation in $\Delta_i$ from a regime in which there are no short-distance singularities. The goal of this section is however to show that the R-product (2.1) can also be determined purely from its analyticity properties and from conformal symmetry. First, Lorentz symmetry implies that it must be a function of the only invariant, $k^2$. Then, scale symmetry determines the overall scaling dimension of the 2-point function in terms of $\Delta_1$ and $\Delta_2$, and we must conclude that

$$\langle\langle R[\phi_1(k), \phi_2(-k)]\rangle\rangle = \mathcal{N} \, (-k^2)^{(\Delta_1+\Delta_2-d)/2}, \tag{2.10}$$

where $\mathcal{N}$ is a constant parameter. Note that for any $k$ in the forward tube it is possible to choose a Lorentz frame in which $\mathrm{im}(k) = (\varepsilon, \mathbf{0})$ with $\varepsilon > 0$. In this frame, either $\mathrm{re}(k^0) \neq 0$ and $k^2$ has a non-zero imaginary part, or $\mathrm{re}(k^0) = 0$ in which case $k^2 = -\varepsilon^2 - \mathbf{k}^2 < 0$. Therefore $-k^2$ is non-negative and it can be raised to non-integer powers using the principal value definition of the logarithm. Eq. (2.10) is therefore the most general ansatz compatible with Poincaré and scale symmetry, and analytic in the forward tube. When acting on this ansatz, the Ward identity (2.3) gives the additional condition $\Delta_1 = \Delta_2$: it is in fact a well-known property of conformal field theory that the only non-trivial 2-point functions involve operators with identical scaling dimensions. In fact, it is usually possible to choose a basis of primary operators such that the only non-zero 2-point functions are those involving identical operators. When this is the case, we write

$$\langle\langle R[\phi(k), \phi(-k)]\rangle\rangle = \mathcal{N} \, (-k^2)^{\Delta-d/2}. \tag{2.11}$$

In conclusion, conformal symmetry fixes the form of the 2-point R-product on its domain of analyticity up to a single coefficient $\mathcal{N}$.

Real momenta lie at the boundary of the forward tube and can be attained in the limit $\varepsilon \to 0_+$. When $k$ is space-like, this gives (using the notation of section 1.1)

$$\langle\langle R[\phi(q), \phi(-q)]\rangle\rangle = \mathcal{N} \, (-q^2)^{\Delta-d/2}. \tag{2.12}$$

---

[3]Ambiguities associated with derivatives of the $\theta$-function at $x = 0$ are resolved by the contact terms.

When $k$ is instead time-like, we get a phase that depends whether it points in the forward or backward direction,

$$\langle\langle R[\phi(\mp p), \phi(\pm p)]\rangle\rangle = \mathcal{N} \, e^{\pm i\pi(\Delta - d/2)} (p^2)^{\Delta - d/2}. \tag{2.13}$$

## 2.2 The Wightman function

The R-product is not symmetric under the exchange of the two points, but it satisfies nevertheless an interesting commutation relation,

$$\langle 0| \, \mathrm{R}[\widetilde{\phi}(x_1), \widetilde{\phi}(x_2)] \, |0\rangle - \langle 0| \, \mathrm{R}[\widetilde{\phi}(x_2), \widetilde{\phi}(x_1)] \, |0\rangle = \langle 0| \, [\widetilde{\phi}(x_1), \widetilde{\phi}(x_2)] \, |0\rangle \,, \tag{2.14}$$

following from eq. (2.5) with the identity $\theta(a) + \theta(-a) = 1$.[4] When Fourier transformed, this identity becomes

$$\langle\langle \mathrm{R}[\phi(k), \phi(-k)]\rangle\rangle - \langle\langle \mathrm{R}[\phi(-k), \phi(k)]\rangle\rangle = \langle\langle \phi(k)\phi(-k)\rangle\rangle - \langle\langle \phi(-k)\phi(k)\rangle\rangle, \tag{2.15}$$

where the correlators on the right-hand side are Wightman functions. One should be careful when using this identity as the two correlators on the left-hand side do not have the same domain of analyticity. However, it certainly applies to real points that lie at the boundary of both domains.[5] When $k$ is real and space-like both Wightman functions on the right-hand side of eq. (2.15) vanish by the spectral condition

$$\langle\langle \phi(-k)\phi(k)\rangle\rangle = 0 \qquad \forall \ k \in \mathbb{R}^{1,d-1} \text{ with } k^0 < |\mathbf{k}| \,. \tag{2.16}$$

We have therefore

$$\langle\langle R[\phi(q), \phi(-q)]\rangle\rangle = \langle\langle R[\phi(-q), \phi(q)]\rangle\rangle, \tag{2.17}$$

in agreement with eq. (2.12). If on the contrary $k$ is time-like, then one of the Wightman functions in eq. (2.15) does not vanish (either one depending whether $k$ is forward- or backward-directed). This can then be used to write

$$\begin{aligned}\langle\langle \phi(-p)\phi(p)\rangle\rangle &= \langle\langle \mathrm{R}[\phi(-p), \phi(p)]\rangle\rangle - \langle\langle \mathrm{R}[\phi(p), \phi(-p)]\rangle\rangle \\ &= \mathcal{N} \, 2i \sin\left[\pi\left(\Delta - \tfrac{d}{2}\right)\right] (p^2)^{\Delta - d/2}.\end{aligned} \tag{2.18}$$

This result can be compared with the spectral representation for the Wightman 2-point function: in any quantum field theory,[6] we must have

$$\langle\langle \phi(-p)\phi(p)\rangle\rangle = 2\pi \int\limits_0^\infty dm^2 \rho(m^2)\theta(p^0)\delta(p^2 - m^2), \tag{2.19}$$

in terms of a non-negative spectral density $\rho(m^2) \geq 0$. In a conformal field theory, this spectral density is a power law, $\rho(m^2) \propto (m^2)^{\Delta - d/2}$, and thus

$$\langle\langle \phi(-p)\phi(p)\rangle\rangle = C_\phi (p^2)^{\Delta - d/2}. \tag{2.20}$$

---

[4] The contact terms are necessarily symmetric under $x_1 \leftrightarrow x_2$ and cancel in this equation.

[5] The two forward tubes only interact at their boundaries, but the true domain of analyticity in $k$ extends beyond the forward tube. For instance, real points with space-like $k$ are well within the domain of analyticity of either of the two correlators.

[6] See ref. [15] for a recent exposition of the spectral representation including modern applications.

The coefficient $C_\phi > 0$ is arbitrary as it defines the normalization of the operator $\phi$. In CFT, however, there exists a standard normalization such that in position space

$$\langle 0| \widetilde{\phi}(x)\widetilde{\phi}(0) |0\rangle = \frac{1}{[-(x^0 + i\varepsilon)^2 + \mathbf{x}^2]^\Delta}. \tag{2.21}$$

In this case, performing the Fourier integral shows that

$$C_\phi = \frac{(4\pi)^{d/2+1}}{2^{2\Delta+1}\Gamma(\Delta)\Gamma\left(\Delta - \frac{d}{2} + 1\right)}. \tag{2.22}$$

Note that this coefficient is positive as long as the unitarity bound is satisfied ($\Delta > \frac{d}{2} - 1$). In the limit $\Delta \to \frac{d}{2} - 1$ it vanishes and the spectral density approaches the free field theory case, $\rho(m^2) \propto \delta(m^2)$.

The compatibility of our result (2.18) with the spectral representation for the Wightman function implies that

$$\mathcal{N} = \frac{C_\phi}{2i\sin\left[\pi\left(\Delta - \frac{d}{2}\right)\right]} = i\,\frac{(4\pi)^{d/2}\Gamma\left(\frac{d}{2} - \Delta\right)}{2^{2\Delta}\Gamma(\Delta)}. \tag{2.23}$$

This shows that the constant $\mathcal{N}$ diverges whenever $\Delta = \frac{d}{2} + n$ with integer $n$. The problem in this case is that the Fourier transform of the position-space R-product does not exist, at least not without regularization: while the Wightman function (2.21) is a tempered distribution admitting a Fourier transform for any $\Delta$, this is not true of the retarded commutator, as its definition involves a $\theta$-function which is not itself a distribution. In fact, the scaling dimensions $\Delta = \frac{d}{2} + n$ correspond precisely to the cases in which contact terms are allowed by conformal symmetry.[7] The Fourier transform of such contact terms are polynomials in $k^2$ that regularize the R-product. Our ability to determine the R-product for any $\Delta \neq \frac{d}{2} + n$ suggests how to proceed by analytic continuation in $\Delta$, without having to construct contact terms explicitly: after renormalization, i.e. after subtracting an appropriate polynomial term proportional to $(-k^2)^n$ on the right-hand side of eq. (2.11), the R-product in these special cases ends up proportional to a logarithm,

$$\langle\!\langle R[\phi(k), \phi(-k)]\rangle\!\rangle \propto (-k^2)^n \log(-k^2). \tag{2.24}$$

## 2.3 The time-ordered product and Schwinger function

The time-ordered 2-point function could be constructed from the spectral density $\rho(m^2)$ using the Källen-Lehmann representation. However, we shall take here a different approach using a simple identity between correlation functions, as this is similar in spirit to our later treatment of the 3-point function. Using the definition of the time-ordered correlator in position space,

$$\begin{aligned}
\langle 0| \mathrm{T}[\widetilde{\phi}_1(x_1)\widetilde{\phi}_2(x_2)] |0\rangle = {}& \theta(x_1^0 - x_2^0)\,\langle 0| \widetilde{\phi}_1(x_1)\widetilde{\phi}_2(x_2) |0\rangle \\
& + \theta(x_2^0 - x_1^0)\,\langle 0| \widetilde{\phi}_2(x_2)\widetilde{\phi}_1(x_1) |0\rangle + \text{contact terms},
\end{aligned} \tag{2.25}$$

---

[7]See ref. [16] for a discussion of the transformation properties of delta functions (and derivative thereof) under conformal symmetry.

it is straightforward to verify the identity

$$\langle 0| \, T[\widetilde{\phi}_1(x_1)\widetilde{\phi}_2(x_2)] \, |0\rangle = \langle 0| \, R[\widetilde{\phi}_1(x_1), \widetilde{\phi}_2(x_2)] \, |0\rangle + \langle 0| \, \widetilde{\phi}_2(x_2)\widetilde{\phi}_1(x_1) \, |0\rangle \, . \qquad (2.26)$$

They are no contact terms in this last equation: if the R-product is regularized to be a tempered distribution, then eq. (2.26) provides a definition of the T-product as a sum of two tempered distributions. After Fourier transform, we obtain therefore

$$\langle\!\langle T[\phi(k)\phi(-k)] \rangle\!\rangle = \langle\!\langle R[\phi(k), \phi(-k)] \rangle\!\rangle + \langle\!\langle \phi(-k)\phi(k) \rangle\!\rangle. \qquad (2.27)$$

We shall take this equation as the definition of the 2-point T-product. Using the R-product and Wightman correlators computed previously, we obtain immediately the time-ordered 2-point function both for space-like momentum,

$$\langle\!\langle T[\phi(q)\phi(-q)] \rangle\!\rangle = \mathcal{N}(-q^2)^{\Delta-d/2} \qquad (2.28)$$

and for time-like momentum,

$$\langle\!\langle T[\phi(\mp p)\phi(\pm p)] \rangle\!\rangle = \mathcal{N}e^{i\pi(\Delta-d/2)}(p^2)^{\Delta-d/2}. \qquad (2.29)$$

As expected, the time-ordered product is symmetric under $p \leftrightarrow -p$.

A peculiarity of this time-ordered 2-point function is that all real kinematic configurations are covered by

$$\langle\!\langle T[\phi(k)\phi(-k)] \rangle\!\rangle = \mathcal{N}(-k^2 + i\varepsilon)^{\Delta-d/2}, \qquad (2.30)$$

in the limit $\varepsilon \to 0_+$. This suggests the possibility of performing a Wick rotation towards Euclidean space, taking $k^0 \to -ik_E^d$ and $\mathbf{k} \to \mathbf{k}_E$ with real $k_E$, without encountering any branch point. With a factor of $-i$ that can be traced back to Wick rotation of the Fourier integral, this gives

$$\langle\!\langle \phi(k_E)\phi(-k_E) \rangle\!\rangle_E = \frac{(4\pi)^{d/2}\Gamma\left(\frac{d}{2} - \Delta\right)}{2^{2\Delta}\Gamma(\Delta)}(k_E^2)^{\Delta-d/2}, \qquad (2.31)$$

which is nothing but the Fourier transform of the Euclidean 2-point function

$$\langle \widetilde{\phi}(x)\widetilde{\phi}(0) \rangle_E = \frac{1}{(x^2)^\Delta}. \qquad (2.32)$$

This concludes the study of 2-point correlation functions. At this stage the whole exercise might seem futile: writing down the different orderings in position space and computing the Fourier transform explicitly might have been an easier task. However, it is quite instructive in the perspective of attacking the more difficult problem of the 3-point function. The key element of the derivation is the construction of the R-product as an analytic function from which the various orderings in Minkowski space can be obtained as boundary values. This is precisely what we will examine next.

# 3 The 3-point R-product

We approach the 3-point function as we did with the 2-point function, starting with the R-product that defines an analytic function in some domain. We will first list the properties that define the 3-point R-product in momentum space, and show how they are motivated by the position-space representation. Then we will construct the most general function consistent with the analyticity properties of the R-product and with conformal symmetry. We will proceed step by step, defining the 3-point function with all space-like momenta first, and then performing successive analytic continuation until the function is defined for all real momenta, space-like or time-like. We shall then find that it is uniquely determined up to a multiplicative factor corresponding to the OPE coefficient.

## 3.1 Definition and permutation symmetry

The 3-point R-product is a function of two momenta, say $k_1$ and $k_2$, that we will denote

$$\langle\!\langle \mathrm{R}[\phi_1(k_1), \phi_2(k_2)\phi_3(k_3)]\rangle\!\rangle, \qquad k_3 = -k_1 - k_2. \tag{3.1}$$

By assumption, this is an analytic function of $k_1$ over the forward tube. We shall take $k_2$ to be real (space-like or time-like), which means that $k_3$ is generically complex and contained in the backward tube. Besides conformal symmetry, which is discussed in details in the next section, the R-product enjoys the permutation symmetry

$$\langle\!\langle \mathrm{R}[\phi_1(k_1), \phi_2(k_2)\phi_3(k_3)]\rangle\!\rangle = \langle\!\langle \mathrm{R}[\phi_1(k_1), \phi_3(k_3)\phi_2(k_2)]\rangle\!\rangle, \tag{3.2}$$

as well as

$$\langle\!\langle \mathrm{R}[\phi_1(k_1), \phi_2(k_2)\phi_3(k_3)]\rangle\!\rangle - \langle\!\langle \mathrm{R}[\phi_2(k_2), \phi_1(k_1)\phi_3(k_3)]\rangle\!\rangle$$
$$= \langle\!\langle \phi_1(k_1)\, \mathrm{R}[\phi_2(k_2), \phi_3(k_3)]\rangle\!\rangle - \langle\!\langle \mathrm{R}[\phi_2(k_2), \phi_3(k_3)]\phi_1(k_1)\rangle\!\rangle$$
$$- \langle\!\langle \phi_2(k_2)\, \mathrm{R}[\phi_1(k_1), \phi_3(k_3)]\rangle\!\rangle + \langle\!\langle \mathrm{R}[\phi_1(k_1), \phi_3(k_3)]\phi_2(k_2)\rangle\!\rangle. \tag{3.3}$$

These properties can be traced back to the definition of the 3-point R-product in position space [17],

$$\langle 0| \mathrm{R}[\widetilde{\phi}_1(x_1), \widetilde{\phi}_2(x_2)\widetilde{\phi}_3(x_3)] |0\rangle = \theta(x_1^0 - x_2^0)\theta(x_2^0 - x_3^0) \langle 0| \left[ [\widetilde{\phi}_1(x_1), \widetilde{\phi}_2(x_2)], \widetilde{\phi}_3(x_3) \right] |0\rangle$$
$$+ \theta(x_1^0 - x_3^0)\theta(x_3^0 - x_2^0) \langle 0| \left[ [\widetilde{\phi}_1(x_1), \widetilde{\phi}_3(x_3)], \widetilde{\phi}_2(x_2) \right] |0\rangle$$
$$+ \text{contact terms.} \tag{3.4}$$

The contact terms include local and semi-local terms that are needed to regularize the singularities when two or more points coincide. By its very definition, the R-product is symmetric under the exchange $2 \leftrightarrow 3$,

$$\langle 0| \mathrm{R}[\widetilde{\phi}_1(x_1), \widetilde{\phi}_2(x_2)\widetilde{\phi}_3(x_3)] |0\rangle = \langle 0| \mathrm{R}[\widetilde{\phi}_1(x_1), \widetilde{\phi}_3(x_3)\widetilde{\phi}_2(x_2)] |0\rangle. \tag{3.5}$$

It is not symmetric under $1 \leftrightarrow 2$, but there exists a relation similar to eq. (2.14) relating the difference of orderings to retarded commutators [18],

$$\langle 0| \mathrm{R}[\widetilde{\phi}_1(x_1), \widetilde{\phi}_2(x_2)\widetilde{\phi}_3(x_3)] |0\rangle - \langle 0| \mathrm{R}[\widetilde{\phi}_2(x_2), \widetilde{\phi}_1(x_1)\widetilde{\phi}_3(x_3)] |0\rangle$$
$$= \theta(x_2^0 - x_3^0) \langle 0| \left[ \widetilde{\phi}_1(x_1), [\widetilde{\phi}_2(x_2), \widetilde{\phi}_3(x_3)] \right] |0\rangle - \theta(x_1^0 - x_3^0) \langle 0| \left[ \widetilde{\phi}_2(x_2), [\widetilde{\phi}_1(x_1), \widetilde{\phi}_3(x_3)] \right] |0\rangle. \tag{3.6}$$

But the specificity of the R-product (3.4) is the retardation property in $x_1$, namely the fact that it vanishes unless $x_1$ sits inside the future light cones of both $x_2$ and $x_3$. As before, this implies that the Fourier transform with respect to $x_1$ yields a function of the momentum $k_1$ that is analytic in the forward tube. This is true independently of the positions $x_2$ and $x_3$, and so they can be Fourier transformed too. We will therefore identify the correlator (3.1) with the Fourier transform of eq. (3.4),

$$\langle\!\langle \mathrm{R}[\phi_1(k_1), \phi_2(k_2)\phi_3(k_3)]\rangle\!\rangle = \int d^d x_1 d^d x_2 \, e^{i(p_1\cdot x_1 + p_2\cdot x_2)} \langle 0| \, \mathrm{R}[\widetilde{\phi}_1(x_1), \widetilde{\phi}_2(x_2)\widetilde{\phi}_3(0)] \, |0\rangle \,. \quad (3.7)$$

The permutation symmetries (3.2) and (3.3) of the momentum-space function are straight-forwardly inherited from eqs. (3.5) and (3.6).

The identity (3.3) is particularly interesting when combined with the spectral condition for Wightman functions, stating that

$$\langle\!\langle \phi(-k)\cdots \rangle\!\rangle = \langle\!\langle \cdots \phi(k) \rangle\!\rangle = 0 \qquad \forall \, k \in \mathbb{R}^{1,d-1} \text{ with } k^0 < |\mathbf{k}| \,, \quad (3.8)$$

i.e. that the eigenstate $\phi(k)|0\rangle$ (or its Hermitian conjugate) has momentum inside the forward light cone. When both $k_1$ and $k_2$ are space-like (hence denoted by $q_1$ and $q_2$ below), all correlators to the right-hand side of eq. (3.3) vanish by the spectral condition and we obtain the identity

$$\langle\!\langle \mathrm{R}[\phi_1(q_1), \phi_2(q_2)\phi_3(k_3)]\rangle\!\rangle = \langle\!\langle \mathrm{R}[\phi_2(q_2), \phi_1(q_1)\phi_3(k_3)]\rangle\!\rangle. \quad (3.9)$$

When combined with conformal symmetry, the permutation identities (3.2) and (3.9) will be sufficient to determine the 3-point R-product completely.

## 3.2 A conformally-symmetric ansatz

Let us now construct the most general ansatz for the R-product that is consistent with conformal symmetry in $d$-dimensional Minkowski space-time. The requirement of Poincaré symmetry is easy to implement: translation symmetry is already used to its full extent in the definition (3.7), and Lorentz symmetry implies that the 3-point correlator is a function of the three invariants $k_1^2$, $k_2^2$ and $k_3^2 = (k_1 + k_2)^2$. The Ward identity for scale transformations is

$$\left(-k_1^\mu \frac{\partial}{\partial k_1^\mu} - k_2^\mu \frac{\partial}{\partial k_2^\mu} + \Delta_1 + \Delta_2 + \Delta_3 - 2d\right) \langle\!\langle \mathrm{R}[\phi_1(k_1), \phi_2(k_2)\phi_3(k_3)]\rangle\!\rangle = 0. \quad (3.10)$$

Like the Ward identity associated with Lorentz transformations, this is a first-order differential equation and therefore it is solved by the most general function of the scale-invariant combination of momenta, which we can take to be $k_2^2/k_1^2$ and $k_3^2/k_1^2$. Since eq. (3.10) is non-homogeneous, we must have

$$\langle\!\langle \mathrm{R}[\phi_1(k_1), \phi_2(k_2)\phi_3(k_3)]\rangle\!\rangle = (-k_1^2)^{(\Delta_1 + \Delta_2 + \Delta_3 - 2d)/2} F\left(\frac{k_2^2}{k_1^2}, \frac{k_3^2}{k_1^2}\right), \quad (3.11)$$

for some unknown function $F$ of two complex variables. Note that $k_1^2$ is never zero or real-positive for all $k_1$ in the forward tube, which means that our ansatz is well-defined

over the whole domain of analyticity of the R-product. Once again, this is most easily seen in the Lorentz frame in which $\text{im}(k_1) = (\varepsilon, \mathbf{0})$.

The constraint imposed by symmetry under special conformal transformations is more complicated: the associated Ward identity is a second-order differential equation analogous to eq. (2.3),

$$\sum_{i=1}^{2} \left( -2k_i^\nu \frac{\partial^2}{\partial k_{i\mu} \partial k_i^\nu} + k_i^\mu \frac{\partial^2}{\partial k_{i\nu} \partial k_i^\nu} + 2(\Delta_i - d)\frac{\partial}{\partial k_{i\mu}} \right) \langle\!\langle R[\phi_1(k_1), \phi_2(k_2)\phi_3(k_3)] \rangle\!\rangle = 0.$$
(3.12)

When applied to the ansatz (3.11), this becomes a vector equation for the function $F$ whose components along $k_1^\mu$ and $k_2^\mu$ are respectively

$$\left[ x(1-x)\frac{\partial^2}{\partial x^2} - 2xy\frac{\partial^2}{\partial x \partial y} - y^2 \frac{\partial^2}{\partial y^2} \right.$$
$$\left. + (\gamma - (\alpha + \beta + 1))\frac{\partial}{\partial x} - (\alpha + \beta + 1)\frac{\partial}{\partial y} - \alpha\beta \right] F(x, y) = 0, \qquad (3.13)$$
$$\left[ y(1-y)\frac{\partial^2}{\partial y^2} - 2xy\frac{\partial^2}{\partial x \partial y} - x^2 \frac{\partial^2}{\partial x^2} \right.$$
$$\left. + (\gamma' - (\alpha + \beta + 1))\frac{\partial}{\partial y} - (\alpha + \beta + 1)\frac{\partial}{\partial x} - \alpha\beta \right] F(x, y) = 0. \qquad (3.14)$$

For simplicity of notation we have denoted the variables $(k_2^2/k_1^2, k_3^2/k_1^2)$ by $(x, y)$, and defined

$$\alpha = \frac{2d - \Delta_1 - \Delta_2 - \Delta_3}{2}, \qquad \gamma = \frac{d}{2} - \Delta_2 + 1,$$
$$\beta = \frac{d + \Delta_1 - \Delta_2 - \Delta_3}{2}, \qquad \gamma' = \frac{d}{2} - \Delta_3 + 1. \qquad (3.15)$$

This system of partial differential equations is solved by the Appell $F_4$ double hypergeometric series [19]

$$F_4\left(\alpha, \beta, \gamma, \gamma'; x, y\right) = \sum_{i,j=0}^{\infty} \frac{(\alpha)_{i+j}(\beta)_{i+j}}{i! j! (\gamma)_i (\gamma')_j} x^i y^j. \qquad (3.16)$$

But for generic values of the scaling dimensions $\Delta_i$, there are in fact four independent solutions given by

$$F_4\left(\frac{2d - \Delta_2 - \Delta_3 - \Delta_1}{2}, \frac{d - \Delta_2 - \Delta_3 + \Delta_1}{2}, \frac{d}{2} - \Delta_2 + 1, \frac{d}{2} - \Delta_3 + 1; x, y\right), \qquad (3.17)$$

$$x^{\Delta_2 - d/2} F_4\left(\frac{d + \Delta_2 - \Delta_3 - \Delta_1}{2}, \frac{\Delta_2 - \Delta_3 + \Delta_1}{2}, \Delta_2 - \frac{d}{2} + 1, \frac{d}{2} - \Delta_3 + 1; x, y\right), \qquad (3.18)$$

$$y^{\Delta_3 - d/2} F_4\left(\frac{d + \Delta_3 - \Delta_2 - \Delta_1}{2}, \frac{\Delta_3 - \Delta_2 + \Delta_1}{2}, \frac{d}{2} - \Delta_2 + 1, \Delta_3 - \frac{d}{2} + 1; x, y\right), \qquad (3.19)$$

and

$$x^{\Delta_2 - d/2} y^{\Delta_3 - d/2} F_4\left(\frac{\Delta_2 + \Delta_3 - \Delta_1}{2}, \frac{\Delta_2 + \Delta_3 + \Delta_1 - d}{2}, \Delta_2 - \frac{d}{2} + 1, \Delta_3 - \frac{d}{2} + 1; x, y\right). \qquad (3.20)$$

The most general solution in a neighborhood of $(x, y) = (0, 0)$ is a linear combination of these four functions.

Note that for generic scaling dimensions of the operators the solutions (3.18) to (3.20) have singularities at $x = 0$ or $y = 0$, corresponding to the momenta $k_2$ or $k_3$ being light-like. Moreover, the series (3.16) only converges when $\sqrt{|x|} + \sqrt{|y|} < 1$: the function that it defines by analytic continuation is singular whenever $1 + x^2 + y^2 - 2x - 2y - 2xy = 0$. In terms of the momenta $k_1$ and $k_2$, this singular locus corresponds to the condition $(k_1 \cdot k_2)^2 = k_1^2 k_2^2$, which is only satisfied if $k_1$ and $k_2$ are colinear vectors. Therefore we need to take great care in relating the 3-point R-product to this basis of solutions. We will adopt the following strategy: first, we make an ansatz for the correlator in a regime where all four solutions (3.17) to (3.20) are non-singular; then we proceed to extend the definition of the correlator by analytic continuation; finally, we impose the symmetry constraints (3.2) and (3.9) to fix the free parameters of our original ansatz.

For our ansatz, we choose to work in the regime of real and space-like momenta satisfying $q_i^2 = -\mu_i^2 < 0$ with $|\mu_2| + |\mu_3| < |\mu_1|$. In this case we have $0 < x, y < 1$ with $\sqrt{|x|} + \sqrt{|y|} < 1$, so that both the $F_4$ hypergeometric series and the non-integer powers of $x$ and $y$ are unambiguously defined. We introduce the function

$$G_{\Delta_a \Delta_b, \Delta_c}(x, y) = \Gamma\left(\frac{d}{2} - \Delta_a\right) \Gamma\left(\frac{d}{2} - \Delta_b\right) \Gamma\left(\frac{\Delta_a + \Delta_b - \Delta_c}{2}\right) \Gamma\left(\frac{\Delta_a + \Delta_b + \Delta_c - d}{2}\right)$$
$$\times F_4\left(\frac{\Delta_a + \Delta_b - \Delta_c}{2}, \frac{\Delta_a + \Delta_b + \Delta_c - d}{2}, \Delta_a - \frac{d}{2} + 1, \Delta_b - \frac{d}{2} + 1; x, y\right),$$
$$(3.21)$$

and write the 3-point R-product as

$$\langle\!\langle R[\phi_1(q_1), \phi_2(q_2)\phi_3(q_3)]\rangle\!\rangle$$
$$= C^{(1)}_{\Delta_1 \Delta_2 \Delta_3}(-q_1^2)^{(\Delta_1 - \Delta_2 - \Delta_3)/2}(-q_2^2)^{\Delta_2 - d/2}(-q_3^2)^{\Delta_3 - d/2} G_{\Delta_2 \Delta_3, \Delta_1}\left(\frac{q_2^2}{q_1^2}, \frac{q_3^2}{q_1^2}\right)$$
$$+ C^{(2)}_{\Delta_1 \Delta_2 \Delta_3}(-q_1^2)^{(\Delta_1 - \Delta_2 + \Delta_3 - d)/2}(-q_2^2)^{\Delta_2 - d/2} G_{\Delta_2 \widetilde{\Delta}_3, \Delta_1}\left(\frac{q_2^2}{q_1^2}, \frac{q_3^2}{q_1^2}\right)$$
$$+ C^{(3)}_{\Delta_1 \Delta_2 \Delta_3}(-q_1^2)^{(\Delta_1 + \Delta_2 - \Delta_3 - d)/2}(-q_3^2)^{\Delta_3 - d/2} G_{\widetilde{\Delta}_2 \Delta_3, \Delta_1}\left(\frac{q_2^2}{q_1^2}, \frac{q_3^2}{q_1^2}\right)$$
$$+ C^{(4)}_{\Delta_1 \Delta_2 \Delta_3}(-q_1^2)^{(\Delta_1 + \Delta_2 + \Delta_3 - 2d)/2} G_{\widetilde{\Delta}_2 \widetilde{\Delta}_3, \Delta_1}\left(\frac{q_2^2}{q_1^2}, \frac{q_3^2}{q_1^2}\right).$$
$$(3.22)$$

The four terms on the right-hand side correspond to the four solutions (3.17) to (3.20), up to conventional $\Gamma$-functions (their role will be clear later). The parameters $C^{(i)}_{\Delta_1 \Delta_2 \Delta_3}$ are complex numbers, unknown at this stage, which might depend on the scaling dimensions of the operators, as indicated by their indices. We have used the notation $\widetilde{\Delta}_i = d - \Delta_i$, which illustrates the fact that the various solutions are related by the shadow transform [20, 21]. Eq. (3.22) is now the most general ansatz consistent with conformal symmetry for the R-product in the specified kinematic regime.

The permutation symmetry (3.2) has an immediate consequence on this ansatz. Using the property $G_{\Delta_a \Delta_b, \Delta_c}(x, y) = G_{\Delta_b \Delta_a, \Delta_c}(y, x)$ that follows from the definition (3.16) of the Appell $F_4$ function, one can see that the symmetry under $2 \leftrightarrow 3$ requires

$$C^{(1)}_{\Delta_1 \Delta_2 \Delta_3} = C^{(1)}_{\Delta_1 \Delta_3 \Delta_2}, \qquad C^{(2)}_{\Delta_1 \Delta_2 \Delta_3} = C^{(3)}_{\Delta_1 \Delta_3 \Delta_2}, \qquad C^{(4)}_{\Delta_1 \Delta_2 \Delta_3} = C^{(4)}_{\Delta_1 \Delta_3 \Delta_2}. \qquad (3.23)$$

It is not obvious however how to impose the symmetry (3.9) exchanging $1 \leftrightarrow 2$. For this, we first need to extend the domain of validity of the ansatz (3.22).

## 3.3 First analytic continuation

The goal of this section is to use the analyticity of the R-product to extend the domain of validity of the ansatz (3.22) to time-like momenta. We will keep $q_2$ fixed to its real space-like value, and perform an analytic continuation of $q_1$ in the forward tube. Note that in doing so $q_3 = -q_1 - q_2$ gets analytically continued in the backward tube. For $q_1$ in the forward tube, it is always possible to choose a frame in which its imaginary part has no spatial component, and write

$$q_1 = (k_1^0 + i\varepsilon, \mathbf{k}_1), \qquad k_1 \in \mathbb{R}^{1,d-1}, \quad \varepsilon > 0. \qquad (3.24)$$

As suggested by the notation, we will be interested in the limit $\varepsilon \to 0_+$ to recover real momenta, but $\varepsilon$ can in fact take any positive value. Let us also introduce the notation

$$[-k_i^2]_\pm = -(k_i^0 \pm i\varepsilon)^2 + \mathbf{k}_i^2. \qquad (3.25)$$

This quantity either has an imaginary part when $k_i^0 \neq 0$, or it is strictly positive for $k_i^0 = 0$. This means that it can be unequivocally raised to a non-integer power. In the limit $\varepsilon \to 0_+$ we have

$$[-k_i^2]_\pm^\alpha = \begin{cases} (-k_i^2)^\alpha & \text{for } |k_i^0| \leq |\mathbf{k}_i|, \\ e^{-i\pi\alpha}(q_i^2)^\alpha & \text{for } k_i^0 > |\mathbf{k}_i|, \\ e^{i\pi\alpha}(q_i^2)^\alpha & \text{for } k_i^0 < -|\mathbf{k}_i|. \end{cases} \qquad (3.26)$$

Using this notation, the real ansatz (3.22) can be lifted to an analytic function in the forward tube, writing

$$\langle\!\langle \mathrm{R}[\phi_1(k_1), \phi_2(k_2)\phi_3(q_3)]\rangle\!\rangle$$

$$= C^{(1)}_{\Delta_1\Delta_2\Delta_3}[-k_1^2]_+^{(\Delta_1-\Delta_2-\Delta_3)/2}(-q_2^2)^{\Delta_2-d/2}[-k_3^2]_-^{\Delta_3-d/2}G_{\Delta_2\Delta_3,\Delta_1}\left(\frac{-q_2^2}{[-k_1^2]_+}, \frac{[-k_3^2]_-}{[-k_1^2]_+}\right)$$

$$+ C^{(2)}_{\Delta_1\Delta_2\Delta_3}[-k_1^2]_+^{(\Delta_1-\Delta_2+\Delta_3-d)/2}(-q_2^2)^{\Delta_2-d/2}G_{\Delta_2\tilde{\Delta}_3,\Delta_1}\left(\frac{-q_2^2}{[-k_1^2]_+}, \frac{[-k_3^2]_-}{[-k_1^2]_+}\right)$$

$$+ C^{(3)}_{\Delta_1\Delta_2\Delta_3}[-k_1^2]_+^{(\Delta_1+\Delta_2-\Delta_3-d)/2}[-k_3^2]_-^{\Delta_3-d/2}G_{\tilde{\Delta}_2\Delta_3,\Delta_1}\left(\frac{-q_2^2}{[-k_1^2]_+}, \frac{[-k_3^2]_-}{[-k_1^2]_+}\right)$$

$$+ C^{(4)}_{\Delta_1\Delta_2\Delta_3}[-k_1^2]_+^{(\Delta_1+\Delta_2+\Delta_3-2d)/2}G_{\tilde{\Delta}_2\tilde{\Delta}_3,\Delta_1}\left(\frac{-q_2^2}{[-k_1^2]_+}, \frac{[-k_3^2]_-}{[-k_1^2]_+}\right). \qquad (3.27)$$

This is a function of two real vectors $k_1$ and $q_2$ (with $k_3 = -k_1 - q_2$), and of $\varepsilon$ (implicit on the left-hand side). As $\varepsilon \to 0_+$, it coincides with eq. (3.22) provided that $k_1$ and $k_3$ are both space-like. At $\varepsilon > 0$, however, $k_1^0$ and $\mathbf{k}_1$ can now be continued to arbitrary real values without leaving the forward tube.[8] For instance, we can make $k_3$ time-like while

---

[8]As long as $k_1$ and $q_2$ are non-colinear and that $\varepsilon$ is infinitesimal, i.e. $\varepsilon \ll |(k_1 \cdot q_2)^2 - k_1^2 q_2^2|$, the arguments of the Appell $F_4$ functions do not intersect the singular locus; at larger values of $\varepsilon$ there are apparent singularities in (3.27), but we shall see that the coefficients $C^{(i)}_{\Delta_1\Delta_2\Delta_3}$ take precisely the right value for these singularities to cancel.

keeping $k_1$ space-like, then take the limit $\varepsilon \to 0_+$, and we arrive at

$$
\begin{aligned}
&\langle\!\langle R[\phi_1(q_1), \phi_2(q_2)\phi_3(\pm p_3)]\rangle\!\rangle \\
&= C^{(1)}_{\Delta_1\Delta_2\Delta_3}(-q_1^2)^{(\Delta_1-\Delta_2-\Delta_3)/2}(-q_2^2)^{\Delta_2-d/2}(p_3^2)^{\Delta_3-d/2}G_{\Delta_2\Delta_3,\Delta_1}\left(\frac{q_2^2}{q_1^2}, \frac{p_3^2}{q_1^2}\right)e^{\pm i\pi(\Delta_3-d/2)} \\
&\quad + C^{(2)}_{\Delta_1\Delta_2\Delta_3}(-q_1^2)^{(\Delta_1-\Delta_2+\Delta_3-d)/2}(-q_2^2)^{\Delta_2-d/2}G_{\Delta_2\widetilde{\Delta}_3,\Delta_1}\left(\frac{q_2^2}{q_1^2}, \frac{p_3^2}{q_1^2}\right) \\
&\quad + C^{(3)}_{\Delta_1\Delta_2\Delta_3}(-q_1^2)^{(\Delta_1+\Delta_2-\Delta_3-d)/2}(p_3^2)^{\Delta_3-d/2}G_{\widetilde{\Delta}_2\Delta_3,\Delta_1}\left(\frac{q_2^2}{q_1^2}, \frac{p_3^2}{q_1^2}\right)e^{\pm i\pi(\Delta_3-d/2)} \\
&\quad + C^{(4)}_{\Delta_1\Delta_2\Delta_3}(-q_1^2)^{(\Delta_1+\Delta_2+\Delta_3-2d)/2}G_{\widetilde{\Delta}_2\widetilde{\Delta}_3,\Delta_1}\left(\frac{q_2^2}{q_1^2}, \frac{p_3^2}{q_1^2}\right).
\end{aligned}
\tag{3.28}
$$

The sign on the left-hand side of this equation and the corresponding phases on the right-hand side indicate whether the momentum of the operator $\phi_3$ points in the forward or backward direction, following our conventions listed in the introduction.

This correlation function is now well-suited to examine the symmetry property (3.9) corresponding to the permutation $1 \leftrightarrow 2$. To do so, one makes use of the identity

$$
\begin{aligned}
G_{\Delta_a\Delta_b,\Delta_c}(x,y) &= \frac{\sin\left[\pi\frac{\Delta_a-\Delta_b+\Delta_c}{2}\right]}{\sin\left[\pi\left(\frac{d}{2}-\Delta_b\right)\right]}(-y)^{-(\Delta_a+\Delta_b+\Delta_c-d)/2}G_{\Delta_a\Delta_c,\Delta_b}\left(\frac{x}{y}, \frac{1}{y}\right) \\
&\quad + \frac{\sin\left[\pi\frac{\Delta_a-\Delta_b-\Delta_c+d}{2}\right]}{\sin\left[\pi\left(\frac{d}{2}-\Delta_b\right)\right]}(-y)^{-(\Delta_a+\Delta_b-\Delta_c)/2}G_{\Delta_a\widetilde{\Delta}_c,\Delta_b}\left(\frac{x}{y}, \frac{1}{y}\right),
\end{aligned}
\tag{3.29}
$$

which follows from a transformation property of the Appell $F_4$ series. Note that this identity could not have been applied to the ansatz (3.22) directly, as it requires the argument $y$ to lie away from the positive real line. On the contrary, it can be readily applied to eq. (3.28), leading to a different but equivalent representation

$$
\begin{aligned}
&\langle\!\langle R[\phi_1(q_1), \phi_2(q_2)\phi_3(\pm p_3)]\rangle\!\rangle \\
&= \widehat{C}^{(1)\pm}_{\Delta_1\Delta_2\Delta_3}(p_3^2)^{(\Delta_3-\Delta_1-\Delta_2)/2}(-q_1^2)^{\Delta_1-d/2}(-q_2^2)^{\Delta_2-d/2}G_{\Delta_1\Delta_2,\Delta_3}\left(\frac{q_1^2}{p_3^2}, \frac{q_2^2}{p_3^2}\right) \\
&\quad + \widehat{C}^{(2)\pm}_{\Delta_1\Delta_2\Delta_3}(p_3^2)^{(\Delta_3-\Delta_1+\Delta_2-d)/2}(-q_1^2)^{\Delta_1-d/2}G_{\Delta_1\widetilde{\Delta}_2,\Delta_3}\left(\frac{q_1^2}{p_3^2}, \frac{q_2^2}{p_3^2}\right) \\
&\quad + \widehat{C}^{(3)\pm}_{\Delta_1\Delta_2\Delta_3}(p_3^2)^{(\Delta_3+\Delta_1-\Delta_2-d)/2}(-q_2^2)^{\Delta_2-d/2}G_{\widetilde{\Delta}_1\Delta_2,\Delta_3}\left(\frac{q_1^2}{p_3^2}, \frac{q_2^2}{p_3^2}\right) \\
&\quad + \widehat{C}^{(4)\pm}_{\Delta_1\Delta_2\Delta_3}(p_3^2)^{(\Delta_3+\Delta_1+\Delta_2-2d)/2}G_{\widetilde{\Delta}_1\widetilde{\Delta}_2,\Delta_3}\left(\frac{q_1^2}{p_3^2}, \frac{q_2^2}{p_3^2}\right),
\end{aligned}
\tag{3.30}
$$

where the $\widehat{C}^{(i)\pm}_{\Delta_1\Delta_2\Delta_3}$ are linear combinations of the $C^{(i)}_{\Delta_1\Delta_2\Delta_3}$, given by

$$\widehat{C}^{(1)\pm}_{\Delta_1\Delta_2\Delta_3} = e^{\pm i\pi(\Delta_3-d/2)}\frac{\sin\left[\pi\frac{\Delta_1+\Delta_2-\Delta_3}{2}\right]}{\sin\left[\pi\left(\frac{d}{2}-\Delta_3\right)\right]}C^{(1)}_{\Delta_1\Delta_2\Delta_3} + \frac{\sin\left[\pi\frac{d-\Delta_1-\Delta_2-\Delta_3}{2}\right]}{\sin\left[\pi\left(\frac{d}{2}-\Delta_3\right)\right]}C^{(2)}_{\Delta_1\Delta_2\Delta_3},$$

$$\widehat{C}^{(2)\pm}_{\Delta_1\Delta_2\Delta_3} = e^{\pm i\pi(\Delta_3-d/2)}\frac{\sin\left[\pi\frac{d+\Delta_1-\Delta_2-\Delta_3}{2}\right]}{\sin\left[\pi\left(\frac{d}{2}-\Delta_3\right)\right]}C^{(3)}_{\Delta_1\Delta_2\Delta_3} + \frac{\sin\left[\pi\frac{\Delta_2-\Delta_1-\Delta_3}{2}\right]}{\sin\left[\pi\left(\frac{d}{2}-\Delta_3\right)\right]}C^{(4)}_{\Delta_1\Delta_2\Delta_3},$$

$$\widehat{C}^{(3)\pm}_{\Delta_1\Delta_2\Delta_3} = e^{\pm i\pi(\Delta_3-d/2)}\frac{\sin\left[\pi\frac{d-\Delta_1+\Delta_2-\Delta_3}{2}\right]}{\sin\left[\pi\left(\frac{d}{2}-\Delta_3\right)\right]}C^{(1)}_{\Delta_1\Delta_2\Delta_3} + \frac{\sin\left[\pi\frac{\Delta_1-\Delta_2-\Delta_3}{2}\right]}{\sin\left[\pi\left(\frac{d}{2}-\Delta_3\right)\right]}C^{(2)}_{\Delta_1\Delta_2\Delta_3},$$

$$\widehat{C}^{(4)\pm}_{\Delta_1\Delta_2\Delta_3} = e^{\pm i\pi(\Delta_3-d/2)}\frac{\sin\left[\pi\frac{2d-\Delta_1-\Delta_2-\Delta_3}{2}\right]}{\sin\left[\pi\left(\frac{d}{2}-\Delta_3\right)\right]}C^{(3)}_{\Delta_1\Delta_2\Delta_3} + \frac{\sin\left(\pi\frac{\Delta_1+\Delta_2-\Delta_3-d}{2}\right)}{\sin\left[\pi\left(\frac{d}{2}-\Delta_3\right)\right]}C^{(4)}_{\Delta_1\Delta_2\Delta_3}.$$

$$(3.31)$$

In this form, the permutation symmetry $1 \leftrightarrow 2$ is satisfied provided that

$$\widehat{C}^{(1)\pm}_{\Delta_1\Delta_2\Delta_3} = \widehat{C}^{(1)\pm}_{\Delta_2\Delta_1\Delta_3}, \qquad \widehat{C}^{(2)\pm}_{\Delta_1\Delta_2\Delta_3} = \widehat{C}^{(3)\pm}_{\Delta_2\Delta_1\Delta_3}, \qquad \widehat{C}^{(4)\pm}_{\Delta_1\Delta_2\Delta_3} = \widehat{C}^{(4)\pm}_{\Delta_2\Delta_1\Delta_3}. \qquad (3.32)$$

These are 6 equations in total. The conditions on $\widehat{C}^{(1)\pm}_{\Delta_1\Delta_2\Delta_3}$ and $\widehat{C}^{(4)\pm}_{\Delta_1\Delta_2\Delta_3}$ imply that all four coefficients $C^{(i)}_{\Delta_1\Delta_2\Delta_3}$ are symmetric under the exchange of $\Delta_1$ and $\Delta_2$. Then, the condition relating $\widehat{C}^{(2)\pm}_{\Delta_1\Delta_2\Delta_3}$ to $\widehat{C}^{(3)\pm}_{\Delta_2\Delta_1\Delta_3}$ permits to establish that

$$C^{(1)}_{\Delta_1\Delta_2\Delta_3} = C^{(3)}_{\Delta_1\Delta_2\Delta_3} \qquad \text{and} \qquad C^{(2)}_{\Delta_1\Delta_2\Delta_3} = C^{(4)}_{\Delta_1\Delta_2\Delta_3}. \qquad (3.33)$$

When combined with the conditions (3.23), this implies that the $C^{(i)}_{\Delta_1\Delta_2\Delta_3}$ are totally symmetric under the exchange of scaling dimensions $\Delta_i$, and also that they are equal to each other:

$$C^{(1)}_{\Delta_1\Delta_2\Delta_3} = C^{(2)}_{\Delta_1\Delta_2\Delta_3} = C^{(3)}_{\Delta_1\Delta_2\Delta_3} = C^{(4)}_{\Delta_1\Delta_2\Delta_3}. \qquad (3.34)$$

This means that the ansatz (3.22) is uniquely determined up to a single complex number. This number is directly related to the OPE coefficient that carries dynamical information about the theory (see eq. (4.25) below). Writing $C^{(i)}_{\Delta_1\Delta_2\Delta_3} = C$, the linear combinations (3.31) can be simplified and turn into simple phases, so that

$$\langle\!\langle \mathrm{R}[\phi_1(q_1),\phi_2(q_2)\phi_3(\pm p_3)]\rangle\!\rangle$$
$$= C\Bigg[ (p_3^2)^{(\Delta_3-\Delta_1-\Delta_2)/2}(-q_1^2)^{\Delta_1-d/2}(-q_2^2)^{\Delta_2-d/2}G_{\Delta_1\Delta_2,\Delta_3}\left(\frac{q_1^2}{p_3^2},\frac{q_2^2}{p_3^2}\right)e^{\pm i\pi(\Delta_3-\Delta_1-\Delta_2)/2}$$
$$+ (p_3^2)^{(\Delta_3-\Delta_1+\Delta_2-d)/2}(-q_1^2)^{\Delta_1-d/2}G_{\Delta_1\widetilde{\Delta}_2,\Delta_3}\left(\frac{q_1^2}{p_3^2},\frac{q_2^2}{p_3^2}\right)e^{\pm i\pi(\Delta_3-\Delta_1+\Delta_2-d)/2}$$
$$+ (p_3^2)^{(\Delta_3+\Delta_1-\Delta_2-d)/2}(-q_2^2)^{\Delta_2-d/2}G_{\widetilde{\Delta}_1\Delta_2,\Delta_3}\left(\frac{q_1^2}{p_3^2},\frac{q_2^2}{p_3^2}\right)e^{\pm i\pi(\Delta_3+\Delta_1-\Delta_2-d)/2}$$
$$+ (p_3^2)^{(\Delta_3+\Delta_1+\Delta_2-2d)/2}G_{\widetilde{\Delta}_1\widetilde{\Delta}_2,\Delta_3}\left(\frac{q_1^2}{p_3^2},\frac{q_2^2}{p_3^2}\right)e^{\pm i\pi(\Delta_3+\Delta_1+\Delta_2-2d)/2}\Bigg]. \qquad (3.35)$$

Similarly, one can use the analytic continuation (3.27) to determine the 3-point R-

product when the first operator carries a time-like momentum,

$$
\begin{aligned}
\langle\!\langle \mathrm{R}[\phi_1(\pm p_1),&\phi_2(q_2)\phi_3(q_3)]\rangle\!\rangle \\
= C\Bigg[\, &(p_1^2)^{(\Delta_1-\Delta_2-\Delta_3)/2}(-q_2^2)^{\Delta_2-d/2}(-q_3^2)^{\Delta_3-d/2}G_{\Delta_2\Delta_3,\Delta_1}\left(\frac{q_2^2}{p_1^2},\frac{q_3^2}{p_1^2}\right)e^{\mp i\pi(\Delta_1-\Delta_2-\Delta_3)/2} \\
&+ (p_1^2)^{(\Delta_1-\Delta_2+\Delta_3-d)/2}(-q_2^2)^{\Delta_2-d/2}G_{\Delta_2\widetilde{\Delta}_3,\Delta_1}\left(\frac{q_2^2}{p_1^2},\frac{q_3^2}{p_1^2}\right)e^{\mp i\pi(\Delta_1-\Delta_2+\Delta_3-d)/2} \\
&+ (p_1^2)^{(\Delta_1+\Delta_2-\Delta_3-d)/2}(-q_3^2)^{\Delta_3-d/2}G_{\widetilde{\Delta}_2\Delta_3,\Delta_1}\left(\frac{q_2^2}{p_1^2},\frac{q_3^2}{p_1^2}\right)e^{\mp i\pi(\Delta_1+\Delta_2-\Delta_3-d)/2} \\
&+ (p_1^2)^{(\Delta_1+\Delta_2+\Delta_3-2d)/2}G_{\widetilde{\Delta}_2\widetilde{\Delta}_3,\Delta_1}\left(\frac{q_2^2}{p_1^2},\frac{q_3^2}{p_1^2}\right)e^{\mp i\pi(\Delta_1+\Delta_2+\Delta_3-2d)/2}\Bigg]. \quad (3.36)
\end{aligned}
$$

or even when both $\phi_1$ and $\phi_3$ carry time-like momenta,

$$
\begin{aligned}
\langle\!\langle \mathrm{R}[\phi_1(\pm p_1),&\phi_2(q_2)\phi_3(\mp p_3)]\rangle\!\rangle \\
= C\Bigg[\, &(-q_2^2)^{(\Delta_2-\Delta_1-\Delta_3)/2}(p_1^2)^{\Delta_1-d/2}(p_3^2)^{\Delta_3-d/2}G_{\Delta_1\Delta_3,\Delta_2}\left(\frac{p_1^2}{q_2^2},\frac{p_3^2}{q_2^2}\right)e^{\mp i\pi(\Delta_1+\Delta_3-d)/2} \\
&+ (-q_2^2)^{(\Delta_2-\Delta_1+\Delta_3-d)/2}(p_1^2)^{\Delta_1-d/2}G_{\Delta_1\widetilde{\Delta}_3,\Delta_2}\left(\frac{p_1^2}{q_2^2},\frac{p_3^2}{q_2^2}\right)e^{\mp i\pi(\Delta_1-d/2)/2} \\
&+ (-q_2^2)^{(\Delta_2+\Delta_1-\Delta_3-d)/2}(p_3^2)^{\Delta_3-d/2}G_{\widetilde{\Delta}_1\Delta_3,\Delta_2}\left(\frac{p_1^2}{q_2^2},\frac{p_3^2}{q_2^2}\right)e^{\mp i\pi(\Delta_3-d/2)/2} \\
&+ (-q_2^2)^{(\Delta_1+\Delta_2+\Delta_3-2d)/2}G_{\widetilde{\Delta}_1\widetilde{\Delta}_3,\Delta_2}\left(\frac{p_1^2}{q_2^2},\frac{p_3^2}{q_2^2}\right)\Bigg].
\end{aligned}
$$
$$(3.37)$$

Note in that case that the momenta of the operators $\phi_1$ and $\phi_3$ must necessarily point in opposite direction, otherwise it is impossible that their sum is space-like. In computing eq. (3.37) we have again made use of the identity (3.29) so that both arguments of the Appell $F_4$ functions are negative and there are no apparent singularities when $p_1^2 = p_3^2$.

Finally, it should be noted that the analytic continuation (3.27) can also be used to evaluate the correlation function at space like momenta $q_i^2 = -\mu_i^2$ when the condition $|\mu_2| + |\mu_3| < |\mu_1|$ is not satisfied. For instance, to cover the case $|\mu_1| + |\mu_2| < |\mu_3|$, one can use the identity (3.29) directly on eq. (3.27) with $\varepsilon > 0$, and only afterwards take the limit $\varepsilon \to 0_+$ to get

$$
\begin{aligned}
\langle\!\langle \mathrm{R}[\phi_1(q_1),&\phi_2(q_2)\phi_3(q_3)]\rangle\!\rangle \\
= C\Bigg[\, &(-q_3^2)^{(\Delta_3-\Delta_1-\Delta_2)/2}(-q_1^2)^{\Delta_1-d/2}(-q_2^2)^{\Delta_2-d/2}G_{\Delta_1\Delta_2,\Delta_3}\left(\frac{q_1^2}{q_3^2},\frac{q_2^2}{q_3^2}\right) \\
&+ (-q_3^2)^{(\Delta_3+\Delta_1-\Delta_1-d)/2}(-q_1^2)^{\Delta_1-d/2}G_{\Delta_1\widetilde{\Delta}_2,\Delta_3}\left(\frac{q_1^2}{q_3^2},\frac{q_2^2}{q_3^2}\right) \\
&+ (-q_3^2)^{(\Delta_3+\Delta_1-\Delta_2-d)/2}(-q_2^2)^{\Delta_2-d/2}G_{\widetilde{\Delta}_1\Delta_2,\Delta_3}\left(\frac{q_1^2}{q_3^2},\frac{q_2^2}{q_3^2}\right) \\
&+ (-q_3^2)^{(\Delta_3+\Delta_1+\Delta_2-2d)/2}G_{\widetilde{\Delta}_1\widetilde{\Delta}_2,\Delta_3}\left(\frac{q_1^2}{q_3^2},\frac{q_2^2}{q_3^2}\right)\Bigg]. \quad (3.38)
\end{aligned}
$$

This is in fact the same as eq. (3.22) with the labels permuted. This result is intuitive, as the two symmetries (3.2) and (3.9) imply that the R-product evaluated at space-like momenta is totally symmetric under the exchange of operators, but it is actually quite subtle, as the two representations (3.22) and (3.38) are valid in different domains.[9] Our results cover therefore all three cases in which $|q_a| + |q_b| < |q_c|$, where $(a, b, c)$ is any permutation of the points $(1, 2, 3)$. But all-space-like configurations that satisfy the triangle inequality $|q_a| + |q_b| \geq |q_c|$ fall outside the domain of convergence of each of the Appell $F_4$ functions taken individually. Nevertheless, it turns out that the linear combinations of Appell $F_4$ functions appearing in eqs. (3.22) and (3.38) is non-singular for all real-positive arguments, and thus covers all space-like configurations.[10]

## 3.4  Second analytic continuation

Not all real configurations of momenta have been discussed yet: the momentum $q_2$ has so far been kept fixed to its space-like value. As the integral (3.7) does not have obvious analyticity properties in $k_2$, it does not make sense to attempt an analytic continuation in this variable.

One could certainly consider the correlation function $\langle\!\langle R[\phi_1(k_1), \phi_2(\pm p_2)\phi_3(k_3)]\rangle\!\rangle$ as a special case of eq. (3.7) in which the momentum $k_2$ is taken to be time-like, forward- or backward-directed: this is a function that is again analytic in $k_1$ over the forward tube (with $k_3 = -k_1 \mp p_2$ in the backward tube), but it is a different function than the one previously discussed. Nevertheless, the two functions share a boundary value that can be used to determine one from the other: since by symmetry $\langle\!\langle R[\phi_1(k_1), \phi_2(\pm p_2)\phi_3(k_3)]\rangle\!\rangle = \langle\!\langle R[\phi_1(k_1), \phi_3(k_3)\phi_2(\pm p_2)]\rangle\!\rangle$, the case of real and space-like $k_1$ and $k_3$ is already covered by eq. (3.35). Continuing $k_1$ away from the real line can be done as in eq. (3.24), and using the notation (3.25), we get

$$\langle\!\langle R[\phi_1(k_1), \phi_2(k_2)\phi_3(\pm p_3)]\rangle\!\rangle$$
$$= C\Bigg[ (p_3^2)^{(\Delta_3-\Delta_1-\Delta_2)/2}[-k_1^2]_+^{\Delta_1-d/2}[-k_2^2]_-^{\Delta_2-d/2} G_{\Delta_1\Delta_2,\Delta_3}\left(\frac{[-k_1^2]_+}{-p_3^2}, \frac{[-k_2^2]_-}{-p_3^2}\right) e^{\pm i\pi(\Delta_3-\Delta_1-\Delta_2)/2}$$
$$+ (p_3^2)^{(\Delta_3-\Delta_1+\Delta_2-d)/2}[-k_1^2]_+^{\Delta_1-d/2} G_{\Delta_1\widetilde{\Delta}_2,\Delta_3}\left(\frac{[-k_1^2]_+}{-p_3^2}, \frac{[-k_2^2]_-}{-p_3^2}\right) e^{\pm i\pi(\Delta_3-\Delta_1+\Delta_2-d)/2}$$
$$+ (p_3^2)^{(\Delta_3+\Delta_1-\Delta_2-d)/2}[-k_2^2]_-^{\Delta_2-d/2} G_{\widetilde{\Delta}_1\Delta_2,\Delta_3}\left(\frac{[-k_1^2]_+}{-p_3^2}, \frac{[-k_2^2]_-}{-p_3^2}\right) e^{\pm i\pi(\Delta_3+\Delta_1-\Delta_2-d)/2}$$
$$+ (p_3^2)^{(\Delta_3+\Delta_1+\Delta_2-2d)/2} G_{\widetilde{\Delta}_1\widetilde{\Delta}_2,\Delta_3}\left(\frac{[-k_1^2]_+}{-p_3^2}, \frac{[-k_2^2]_-}{-p_3^2}\right) e^{\pm i\pi(\Delta_3+\Delta_1+\Delta_2-2d)/2}\Bigg]. \tag{3.39}$$

It can be checked that this analytic continuation reproduces the result (3.37), and it also

---

[9]The permutation symmetry was presumably used in ref. [13] to argue that the Euclidean 3-point function admits a similar representation, although the discussion there does not mention how to use the identity (3.29) with positive arguments of the Appell $F_4$ function.

[10]This non-trivial fact can be verified using an expansion of the Appell $F_4$ function around the point $(x, y) = (0, 1)$ [19], or alternatively using the representation of the linear combination of functions as an integral over three Bessel $K$ functions [14].

covers new cases, either with two momenta time-like,

$$\langle\langle \mathrm{R}[\phi_1(q_1),\phi_2(\pm p_2)\phi_3(\mp p_3)]\rangle\rangle$$
$$= C\Bigg[ (-q_1^2)^{(\Delta_1-\Delta_2-\Delta_3)/2}(p_2^2)^{\Delta_2-d/2}(p_3^2)^{\Delta_3-d/2}G_{\Delta_2\Delta_3,\Delta_1}\left(\frac{p_2^2}{q_1^2},\frac{p_3^2}{q_1^2}\right)e^{\pm i\pi(\Delta_2-\Delta_3)}$$
$$+ (-q_1^2)^{(\Delta_1-\Delta_2+\Delta_3-d)/2}(p_2^2)^{\Delta_2-d/2}G_{\Delta_2\widetilde{\Delta}_3,\Delta_1}\left(\frac{p_2^2}{q_1^2},\frac{p_3^2}{q_1^2}\right)e^{\pm i\pi(\Delta_2-d/2)}$$
$$+ (-q_1^2)^{(\Delta_1+\Delta_2-\Delta_3-d)/2}(p_3^2)^{\Delta_3-d/2}G_{\widetilde{\Delta}_2\Delta_3,\Delta_1}\left(\frac{p_2^2}{q_1^2},\frac{p_3^2}{q_1^2}\right)e^{\mp i\pi(\Delta_3-d/2)}$$
$$+ (-q_1^2)^{(\Delta_1+\Delta_2+\Delta_3-2d)/2}G_{\widetilde{\Delta}_2\widetilde{\Delta}_3,\Delta_1}\left(\frac{p_2^2}{q_1^2},\frac{p_3^2}{q_1^2}\right)\Bigg], \tag{3.40}$$

or when all three momenta are time-like,

$$\langle\langle \mathrm{R}[\phi_1(\mp p_1),\phi_2(\mp p_2)\phi_3(\pm p_3)]\rangle\rangle$$
$$= C\,e^{\pm i\pi(\Delta_1-\Delta_2+\Delta_3-d)/2}$$
$$\times\Bigg[ (p_3^2)^{(\Delta_3-\Delta_1-\Delta_2)/2}(p_1^2)^{\Delta_1-d/2}(p_2^2)^{\Delta_2-d/2}G_{\Delta_1\Delta_2,\Delta_3}\left(\frac{p_1^2}{p_3^2},\frac{p_2^2}{p_3^2}\right)e^{\mp i\pi(\Delta_2-d/2)}$$
$$+ (p_3^2)^{(\Delta_3+\Delta_1-\Delta_1-d)/2}(p_1^2)^{\Delta_1-d/2}G_{\Delta_1\widetilde{\Delta}_2,\Delta_3}\left(\frac{p_1^2}{p_3^2},\frac{p_2^2}{p_3^2}\right)e^{\pm i\pi(\Delta_2-d/2)}$$
$$+ (p_3^2)^{(\Delta_3+\Delta_1-\Delta_2-d)/2}(p_2^2)^{\Delta_2-d/2}G_{\widetilde{\Delta}_1\Delta_2,\Delta_3}\left(\frac{p_1^2}{p_3^2},\frac{p_2^2}{p_3^2}\right)e^{\mp i\pi(\Delta_2-d/2)}$$
$$+ (p_3^2)^{(\Delta_3+\Delta_1+\Delta_2-2d)/2}G_{\widetilde{\Delta}_1\widetilde{\Delta}_2,\Delta_3}\left(\frac{p_1^2}{p_3^2},\frac{p_2^2}{p_3^2}\right)e^{\pm i\pi(\Delta_2-d/2)}\Bigg] \tag{3.41}$$

and

$$\langle\langle \mathrm{R}[\phi_1(\mp p_1),\phi_2(\pm p_2)\phi_3(\pm p_3)]\rangle\rangle$$
$$= C\,e^{\pm i\pi(\Delta_1+\Delta_2+\Delta_3-2d)/2}$$
$$\times\Bigg[ (p_1^2)^{(\Delta_1-\Delta_2-\Delta_3)/2}(p_2^2)^{\Delta_2-d/2}(p_3^2)^{\Delta_3-d/2}G_{\Delta_2\Delta_3,\Delta_1}\left(\frac{p_2^2}{p_1^2},\frac{p_3^2}{p_1^2}\right)$$
$$+ (p_1^2)^{(\Delta_1-\Delta_2+\Delta_3-d)/2}(p_2^2)^{\Delta_2-d/2}G_{\Delta_2\widetilde{\Delta}_3,\Delta_1}\left(\frac{p_2^2}{p_1^2},\frac{p_3^2}{p_1^2}\right)$$
$$+ (p_1^2)^{(\Delta_1+\Delta_2-\Delta_3-d)/2}(p_3^2)^{\Delta_3-d/2}G_{\widetilde{\Delta}_2\Delta_3,\Delta_1}\left(\frac{p_2^2}{p_1^2},\frac{p_3^2}{p_1^2}\right)$$
$$+ (p_1^2)^{(\Delta_1+\Delta_2+\Delta_3-2d)/2}G_{\widetilde{\Delta}_2\widetilde{\Delta}_3,\Delta_1}\left(\frac{p_2^2}{p_1^2},\frac{p_3^2}{p_1^2}\right)\Bigg]. \tag{3.42}$$

To derive this last equation we have made use of the identity (3.29) before taking the limit $\varepsilon \to 0_+$. Note that when all three momenta are time-like, $p_i^2 = m_i^2 > 0$, with say $p_a$ and $p_b$ pointing in one direction and $p_c$ in the other, then the inequality $|m_a| + |m_b| < |m_c|$ is necessarily satisfied, so that this case always fall in the domain of convergence of the hypergeometric series.

This concludes the study of correlation functions involving the 3-point R-product. All cases of real momenta have been enumerated, and in each case the correlator has been

constructed using successive analytic continuations from the all-space-like configuration to the all-time-like ones. For convenience, we give later in table 1 the complete list of references to the equations covering each individual case. For now, it is not clear whether these R-products will find concrete applications in physical problems. However, we will see in the next section that they are very useful as a tool to construct other types of correlators, from Schwinger to Wightman functions.

# 4  Wightman functions

The construction of Wightman functions follows from the results of the previous section and from the use of identity (3.3) that relates the commutator of 3-point R-products to correlators involving a retarded commutator. We discuss such correlators first, and relate them to pure Wightman functions afterwards.

## 4.1  Retarded commutators

We saw previously how the identity (3.3) could be used to derive the permutation symmetry (3.9), by taking the momenta $k_1$ and $k_2$ to be both space-like, so that all Wightman functions on the right-hand side of eq. (3.3) vanish by the spectral condition. The same logic can be applied to isolate a single Wightman function on the right-hand side of that equation: with $k_2$ space-like and $k_1$ time-like backward-directed, we get the identity

$$\langle\!\langle \mathrm{R}[\phi_1(-p_1), \phi_2(q_2)\phi_3(k_3)]\rangle\!\rangle - \langle\!\langle \mathrm{R}[\phi_2(q_2), \phi_1(-p_1)\phi_3(k_3)]\rangle\!\rangle = \langle\!\langle \phi_1(-p_1)\,\mathrm{R}[\phi_2(q_2), \phi_3(k_3)]\rangle\!\rangle. \tag{4.1}$$

This determines the Wightman function involving a retarded commutator on the right-hand side as the difference of two R-products computed in the previous section. At space-like $k_3$, using the correlators (3.35) and (3.36), we get

$$\langle\!\langle \phi_1(-p_1)\,\mathrm{R}[\phi_2(q_2), \phi_3(q_3)]\rangle\!\rangle$$
$$= 2iC\bigg[(p_1^2)^{(\Delta_1-\Delta_2-\Delta_3)/2}(-q_2^2)^{\Delta_2-d/2}(-q_3^2)^{\Delta_3-d/2}G_{\Delta_2\Delta_3,\Delta_1}\left(\frac{q_2^2}{p_1^2}, \frac{q_3^2}{p_1^2}\right)\sin\left(\pi\tfrac{\Delta_1-\Delta_2-\Delta_3}{2}\right)$$
$$+ (p_1^2)^{(\Delta_1-\Delta_2+\Delta_3-d)/2}(-q_2^2)^{\Delta_2-d/2}G_{\Delta_2\widetilde{\Delta}_3,\Delta_1}\left(\frac{q_2^2}{p_1^2}, \frac{q_3^2}{p_1^2}\right)\sin\left(\pi\tfrac{\Delta_1-\Delta_2+\Delta_3-d}{2}\right)$$
$$+ (p_1^2)^{(\Delta_1+\Delta_2-\Delta_3-d)/2}(-q_3^2)^{\Delta_3-d/2}G_{\widetilde{\Delta}_2\Delta_3,\Delta_1}\left(\frac{q_2^2}{p_1^2}, \frac{q_3^2}{p_1^2}\right)\sin\left(\pi\tfrac{\Delta_1+\Delta_2-\Delta_3-d}{2}\right)$$
$$+ (p_1^2)^{(\Delta_1+\Delta_2+\Delta_3-2d)/2}G_{\widetilde{\Delta}_2\widetilde{\Delta}_3,\Delta_1}\left(\frac{q_2^2}{p_1^2}, \frac{q_3^2}{p_1^2}\right)\sin\left(\pi\tfrac{\Delta_1+\Delta_2+\Delta_3-2d}{2}\right)\bigg]. \tag{4.2}$$

Note that the different phases combine perfectly into sines. This correlation function is in fact more remarkable when it is expressed in a different basis of Appell $F_4$ functions: using the transformation property (3.29), it can be rewritten

$$\langle\!\langle \phi_1(-p_1)\,\mathrm{R}[\phi_2(q_2), \phi_3(q_3)]\rangle\!\rangle = 2iC\sin\left[\pi\left(\Delta_1-\tfrac{d}{2}\right)\right](p_1^2)^{\Delta_1-d/2}$$
$$\times\bigg[(-q_3^2)^{(\Delta_3-\Delta_1-\Delta_2)/2}(-q_2^2)^{\Delta_2-d/2}G_{\Delta_1\Delta_2,\Delta_3}\left(\frac{p_1^2}{q_3^2}, \frac{q_2^2}{q_3^2}\right)$$
$$+ (-q_3^2)^{(\Delta_3-\Delta_1+\Delta_2-d)/2}G_{\Delta_1\widetilde{\Delta}_2,\Delta_3}\left(\frac{p_1^2}{q_3^2}, \frac{q_2^2}{q_3^2}\right)\bigg]. \tag{4.3}$$

Out of the four solutions to the conformal Ward identities, only two appear here. This contrasts with the correlators obtained in the previous section which were all linear combinations of all four solutions, independently of the choice of basis. Now for this Wightman correlation function two of the Appell $F_4$ function cancel out in the difference on the left-hand side of eq. (4.1).

A similar result with the retarded commutator to the left of the third operator is obtained making use of (3.3) with a forward-directed $k_1$. After relabeling the operators for clarity, we find

$$\langle\!\langle \mathrm{R}[\phi_1(q_1), \phi_2(q_2)]\phi_3(p_3)\rangle\!\rangle = 2iC \sin\left[\pi\left(\Delta_3 - \tfrac{d}{2}\right)\right] (p_3^2)^{\Delta_3 - d/2}$$
$$\times \left[ (-q_1^2)^{(\Delta_1 - \Delta_2 - \Delta_3)/2}(-q_2^2)^{\Delta_2 - d/2} G_{\Delta_2\Delta_3,\Delta_1}\left(\frac{q_2^2}{q_1^2}, \frac{p_3^2}{q_1^2}\right) \right.$$
$$\left. + (-q_1^2)^{(\Delta_1 + \Delta_2 - \Delta_3 - d)/2} G_{\widetilde{\Delta}_2\Delta_3,\Delta_1}\left(\frac{q_2^2}{q_1^2}, \frac{p_3^2}{q_1^2}\right) \right]. \quad (4.4)$$

Further results in which more than one momentum is time-like can be obtained using the analyticity properties of this correlation function. Note that it can be written as the Fourier integral

$$\langle\!\langle \mathrm{R}[\phi_1(k_1), \phi_2(k_2)]\phi_3(p_3)\rangle\!\rangle = \int d^d x_1 d^d x_3\, e^{i(k_1 \cdot x_1 + p_3 \cdot x_3)}\langle\!\langle \mathrm{R}[\widetilde{\phi}_1(x_1), \widetilde{\phi}_2(0)]\widetilde{\phi}_3(x_3)\rangle\!\rangle, \quad (4.5)$$

with $k_2 = -k_1 - p_3$, where we have made use of translation symmetry to place the second operator at the origin instead of the third. The position-space correlator in the integrand is defined as

$$\langle\!\langle \mathrm{R}[\widetilde{\phi}_1(x_1), \widetilde{\phi}_2(0)]\widetilde{\phi}_3(x_3)\rangle\!\rangle = \theta(x_1^0)\langle\!\langle [\widetilde{\phi}_1(x_1), \widetilde{\phi}_2(0)]\widetilde{\phi}_3(x_3)\rangle\!\rangle + \text{contact terms.} \quad (4.6)$$

This function vanishes unless $x_1$ is in the future light cone, and therefore its Fourier transform is analytic with respect to $k_1$ in the forward tube, exactly like the 3-point R-product. Making use of the notation (3.25), we can therefore write

$$\langle\!\langle \mathrm{R}[\phi_1(k_1), \phi_2(k_2)]\phi_3(p_3)\rangle\!\rangle$$
$$= 2iC \sin\left[\pi\left(\Delta_3 - \tfrac{d}{2}\right)\right] (p_3^2)^{\Delta_3 - d/2}$$
$$\times \left[ [-k_1^2]_+^{(\Delta_1 - \Delta_2 - \Delta_3)/2}[-k_2^2]_-^{\Delta_2 - d/2} G_{\Delta_2\Delta_3,\Delta_1}\left(\frac{[-k_2^2]_-}{[-k_1^2]_+}, \frac{-p_3^2}{[-k_1^2]_+}\right) \right.$$
$$\left. + [-k_1^2]_+^{(\Delta_1 + \Delta_2 - \Delta_3 - d)/2} G_{\widetilde{\Delta}_2\Delta_3,\Delta_1}\left(\frac{[-k_2^2]_-}{[-k_1^2]_+}, \frac{-p_3^2}{[-k_1^2]_+}\right) \right] \quad (4.7)$$

as well as

$$\langle\!\langle \phi_1(-p_1)\, \mathrm{R}[\phi_2(k_2), \phi_3(k_3)]\rangle\!\rangle$$
$$= 2iC \sin\left[\pi\left(\Delta_1 - \tfrac{d}{2}\right)\right] (p_1^2)^{\Delta_1 - d/2}$$
$$\times \left[ [-k_3^2]_-^{(\Delta_3 - \Delta_1 - \Delta_2)/2}[-k_2^2]_+^{\Delta_2 - d/2} G_{\Delta_1\Delta_2,\Delta_3}\left(\frac{-p_1^2}{[-k_3^2]_-}, \frac{[-k_2^2]_+}{[-k_3^2]_-}\right) \right.$$
$$\left. + [-k_3^2]_-^{(\Delta_3 - \Delta_1 + \Delta_2 - d)/2} G_{\Delta_1\widetilde{\Delta}_2,\Delta_3}\left(\frac{-p_1^2}{[-k_3^2]_-}, \frac{[-k_2^2]_+}{[-k_3^2]_-}\right) \right], \quad (4.8)$$

covering all configurations of real momenta in the limit $\varepsilon \to 0_+$. For completeness, we list next all these configuration explicitly. When either one of the momenta $k_1$ or $k_2$ is time-like (necessarily backward-directed), we have

$$
\begin{aligned}
\langle\!\langle \mathrm{R}[\phi_1(-p_1), &\phi_2(q_2)]\phi_3(p_3)\rangle\!\rangle \\
&= 2iC \sin\left[\pi\left(\Delta_3 - \tfrac{d}{2}\right)\right] (p_3^2)^{\Delta_3 - d/2} \\
&\times \left[ (-q_2^2)^{(\Delta_2 - \Delta_1 - \Delta_3)/2} (p_1^2)^{\Delta_1 - d/2} G_{\Delta_1 \Delta_3, \Delta_2}\left(\frac{p_1^2}{q_2^2}, \frac{p_3^2}{q_2^2}\right) e^{i\pi(\Delta_1 - d/2)} \right. \\
&\left. \quad + (-q_2^2)^{(\Delta_1 + \Delta_2 - \Delta_3 - d)/2} G_{\widetilde{\Delta}_1 \Delta_3, \Delta_2}\left(\frac{p_1^2}{q_2^2}, \frac{p_3^2}{q_2^2}\right) \right].
\end{aligned}
\tag{4.9}
$$

and

$$
\begin{aligned}
\langle\!\langle \mathrm{R}[\phi_1(q_1), &\phi_2(-p_2)]\phi_3(p_3)\rangle\!\rangle \\
&= 2iC \sin\left[\pi\left(\Delta_3 - \tfrac{d}{2}\right)\right] (p_3^2)^{\Delta_3 - d/2} \\
&\times \left[ (-q_1^2)^{(\Delta_1 - \Delta_2 - \Delta_3)/2} (p_2^2)^{\Delta_2 - d/2} G_{\Delta_2 \Delta_3, \Delta_1}\left(\frac{p_2^2}{q_1^2}, \frac{p_3^2}{q_1^2}\right) e^{-i\pi(\Delta_2 - d/2)} \right. \\
&\left. \quad + (-q_1^2)^{(\Delta_1 + \Delta_2 - \Delta_3 - d)/2} G_{\widetilde{\Delta}_2 \Delta_3, \Delta_1}\left(\frac{p_2^2}{q_1^2}, \frac{p_3^2}{q_1^2}\right) \right],
\end{aligned}
\tag{4.10}
$$

choosing in each case the space-like vector to be in the denominator of the arguments of the Appell $F_4$ function. When both $k_1$ and $k_2$ are timelike and pointing in opposite directions, we get

$$
\begin{aligned}
\langle\!\langle \mathrm{R}[\phi_1(p_1), \phi_2(-p_2)]\phi_3(p_3)\rangle\!\rangle &= 2iC e^{i\pi(\Delta_3 - \Delta_1 - \Delta_2 + d)/2} \sin\left[\pi\left(\Delta_3 - \tfrac{d}{2}\right)\right] (p_3^2)^{\Delta_3 - d/2} \\
&\times \left[ (p_2^2)^{(\Delta_2 - \Delta_1 - \Delta_3)/2} (p_1^2)^{\Delta_1 - d/2} G_{\Delta_1 \Delta_3, \Delta_2}\left(\frac{p_1^2}{p_2^2}, \frac{p_3^2}{p_2^2}\right) \right. \\
&\left. \quad + (p_2^2)^{(\Delta_1 + \Delta_2 - \Delta_3 - d)/2} G_{\widetilde{\Delta}_1 \Delta_3, \Delta_2}\left(\frac{p_1^2}{p_2^2}, \frac{p_3^2}{p_2^2}\right) \right],
\end{aligned}
\tag{4.11}
$$

and

$$
\begin{aligned}
\langle\!\langle \mathrm{R}[\phi_1(-p_1), \phi_2(p_2)]\phi_3(p_3)\rangle\!\rangle &= 2iC e^{-i\pi(\Delta_3 - \Delta_1 - \Delta_2 + d)/2} \sin\left[\pi\left(\Delta_3 - \tfrac{d}{2}\right)\right] (p_3^2)^{\Delta_3 - d/2} \\
&\times \left[ (p_1^2)^{(\Delta_1 - \Delta_2 - \Delta_3)/2} (p_2^2)^{\Delta_2 - d/2} G_{\Delta_2 \Delta_3, \Delta_1}\left(\frac{p_2^2}{p_1^2}, \frac{p_3^2}{p_1^2}\right) \right. \\
&\left. \quad + (p_1^2)^{(\Delta_1 + \Delta_2 - \Delta_3 - d)/2} G_{\widetilde{\Delta}_2 \Delta_3, \Delta_1}\left(\frac{p_2^2}{p_1^2}, \frac{p_3^2}{p_1^2}\right) \right].
\end{aligned}
\tag{4.12}
$$

Finally, when they are both time-like and backward directed, we are in a situation in which $p_i^2 = m_i^2$ with $|m_1| + |m_2| < |m_3|$, so that it makes sense to first use the transformation property (3.29) to get in the domain of convergence of the Appell series, after which we

find

$$\langle\!\langle \mathrm{R}[\phi_1(-p_1),\phi_2(-p_2)]\phi_3(p_3)\rangle\!\rangle$$
$$= 2iC\bigg[(p_3^2)^{(\Delta_3-\Delta_1-\Delta_2)/2}(p_1^2)^{\Delta_1-d/2}(p_2^2)^{\Delta_2-d/2}G_{\Delta_1\Delta_2,\Delta_3}\left(\frac{p_1^2}{p_3^2},\frac{p_2^2}{p_3^2}\right)\sin\left(\pi\tfrac{\Delta_3-\Delta_1-\Delta_2}{2}\right)e^{i\pi(\Delta_1-\Delta_2)}$$
$$+ (p_3^2)^{(\Delta_3-\Delta_1+\Delta_2-d)/2}(p_1^2)^{\Delta_1-d/2}G_{\Delta_1\widetilde{\Delta}_2,\Delta_3}\left(\frac{p_1^2}{p_3^2},\frac{p_2^2}{p_3^2}\right)\sin\left(\pi\tfrac{\Delta_3-\Delta_1+\Delta_2-d}{2}\right)e^{i\pi(\Delta_1-d/2)}$$
$$+ (p_3^2)^{(\Delta_3+\Delta_1-\Delta_2-d)/2}(p_2^2)^{\Delta_2-d/2}G_{\widetilde{\Delta}_1\Delta_2,\Delta_3}\left(\frac{p_1^2}{p_3^2},\frac{p_2^2}{p_3^2}\right)\sin\left(\pi\tfrac{\Delta_3+\Delta_1-\Delta_2-d}{2}\right)e^{-i\pi(\Delta_2-d/2)}$$
$$+ (p_3^2)^{(\Delta_3+\Delta_1+\Delta_2-2d)/2}G_{\widetilde{\Delta}_1\widetilde{\Delta}_2,\Delta_3}\left(\frac{p_1^2}{p_3^2},\frac{p_2^2}{p_3^2}\right)\sin\left(\pi\tfrac{\Delta_3+\Delta_1+\Delta_2-2d}{2}\right)\bigg]. \tag{4.13}$$

Similar results valid when the retarded commutator is to the right of the third operator can be obtained by Hermitian conjugation: from the definition (4.6), we have

$$\langle 0|\, \mathrm{R}[\widetilde{\phi}_1(x_1),\widetilde{\phi}_2(x_2)]\widetilde{\phi}_3(x_3)\,|0\rangle^* = -\,\langle 0|\,\widetilde{\phi}_3(x_3)\,\mathrm{R}[\widetilde{\phi}_1(x_1),\widetilde{\phi}_2(x_2)]\,|0\rangle\,, \tag{4.14}$$

which upon Fourier transform gives

$$\langle\!\langle \mathrm{R}[\phi_1(k_1),\phi_2(k_2)]\phi_3(p_3)\rangle\!\rangle^* = -\langle\!\langle \phi_3(-p_3)\,\mathrm{R}[\phi_1(-k_1),\phi_2(-k_2)]\rangle\!\rangle. \tag{4.15}$$

Note that this property can be seen in eqs. (4.7) and (4.8), provided that the OPE coefficient is real, i.e. $C^* = C$. Examining our results, one also observes that for all real momenta

$$\langle\!\langle \mathrm{R}[\phi_1(k_1),\phi_2(k_2)]\phi_3(p_3)\rangle\!\rangle = -\langle\!\langle \mathrm{R}[\phi_2(k_2),\phi_1(k_1)]\phi_3(p_3)\rangle\!\rangle^* \tag{4.16}$$

and

$$\langle\!\langle \phi_1(-p_1)\,\mathrm{R}[\phi_2(k_2),\phi_3(k_3)]\rangle\!\rangle = -\langle\!\langle \phi_1(-p_1)\,\mathrm{R}[\phi_3(k_3),\phi_2(k_2)]\rangle\!\rangle^*. \tag{4.17}$$

As a consequence, we get the relation

$$\langle\!\langle \phi_1(-p_1)\,\mathrm{R}[\phi_2(k_2),\phi_3(k_3)]\rangle\!\rangle = \langle\!\langle \mathrm{R}[\phi_3(-k_3),\phi_2(-k_2)]\phi_1(p_1)\rangle\!\rangle. \tag{4.18}$$

This equation together with the results presented above can be used to verify that the identity (3.3) is satisfied in all cases of real momenta.

This type of 3-point correlators involving a retarded commutator is of physical significance to the momentum-space OPE of a 4-point function involving a double commutator [22], as in the Lorentzian inversion formula [23, 24]. But for us it is also the last intermediate step before obtaining the Wightman 3-point function.

## 4.2 The Wightman 3-point function

The correlator in which the operator are not ordered (i.e. when they act on the vacuum one after the other) can be obtained from the previous results using the identity

$$\langle\!\langle \mathrm{R}[\phi_1(k_1),\phi_2(k_2)]\phi_3(p_3)\rangle\!\rangle - \langle\!\langle \mathrm{R}[\phi_2(k_2),\phi_1(k_1)]\phi_3(p_3)\rangle\!\rangle$$
$$= \langle\!\langle \phi_1(k_1)\phi_2(k_2)\phi_3(p_3)\rangle\!\rangle - \langle\!\langle \phi_2(k_2)\phi_1(k_1)\phi_3(p_3)\rangle\!\rangle. \tag{4.19}$$

This is the generalization of eq. (2.15) to 3-point functions; it follows straightforwardly from a position-space identity similar to eq. (2.14).

When $k_1$ is time-like backward-directed and $k_2$ is space-like, the second term on the right-hand side of this equation vanishes by the spectral condition, and the left-hand side is given by the difference between eqs. (4.9) and (4.10). We get

$$\langle\!\langle \phi_1(-p_1)\phi_2(q_2)\phi_3(p_3)\rangle\!\rangle = -4C \sin\left[\pi\left(\Delta_1 - \tfrac{d}{2}\right)\right]\sin\left[\pi\left(\Delta_3 - \tfrac{d}{2}\right)\right]$$
$$\times (-q_2^2)^{(\Delta_2 - \Delta_1 - \Delta_3)/2}(p_1^2)^{\Delta_1 - d/2}(p_3^2)^{\Delta_3 - d/2}G_{\Delta_1\Delta_3,\Delta_2}\left(\frac{p_1^2}{q_2^2}, \frac{p_3^2}{q_2^2}\right).$$
(4.20)

Strikingly, as first observed in ref. [11], the Wightman 3-point function in this regime can be expressed as a single Appell $F_4$ function: the second $F_4$ function present in eqs. (4.9) and (4.10) precisely cancels out, and the phases multiplying the first $F_4$ function combine into a sine.

Another non-vanishing Wightman function is obtained from the identity (4.19) with $k_1$ and $k_2$ both time-like but pointing in opposite directions: once again, only one term survives on the right-hand side of that identity, and combining eq. (4.11) and (4.12), we get

$$\langle\!\langle \phi_1(-p_1)\phi_2(p_2)\phi_3(p_3)\rangle\!\rangle = 4C \sin\left[\pi\left(\Delta_3 - \tfrac{d}{2}\right)\right]\sin\left[\pi\frac{\Delta_3 - \Delta_1 - \Delta_2 + d}{2}\right](p_3^2)^{\Delta_3 - d/2}$$
$$\times \left[(p_1^2)^{(\Delta_1 - \Delta_2 - \Delta_3)/2}(p_2^2)^{\Delta_2 - d/2}G_{\Delta_2\Delta_3,\Delta_1}\left(\frac{p_2^2}{p_1^2}, \frac{p_3^2}{p_1^2}\right)\right.$$
$$\left. + (p_1^2)^{(\Delta_1 + \Delta_2 - \Delta_3 - d)/2}G_{\widetilde{\Delta}_2\Delta_3,\Delta_1}\left(\frac{p_2^2}{p_1^2}, \frac{p_3^2}{p_1^2}\right)\right].$$
(4.21)

Finally, the result in the case where $k_1$ and $k_2$ are both backward-directed can be obtained from the identity

$$\langle\!\langle \phi_1(-p_1)\,\mathrm{R}[\phi_2(k_2),\phi_3(k_3)]\rangle\!\rangle - \langle\!\langle \phi_1(-p_1)\,\mathrm{R}[\phi_3(k_3),\phi_2(k_2)]\rangle\!\rangle$$
$$= \langle\!\langle \phi_1(-p_1)\phi_2(k_2)\phi_3(k_3)\rangle\!\rangle - \langle\!\langle \phi_1(-p_1)\phi_3(k_3)\phi_2(k_2)\rangle\!\rangle,$$
(4.22)

together with eq. (4.8), and we find

$$\langle\!\langle \phi_1(-p_1)\phi_2(-p_2)\phi_3(p_3)\rangle\!\rangle = 4C \sin\left[\pi\left(\Delta_1 - \tfrac{d}{2}\right)\right]\sin\left[\pi\frac{\Delta_1 - \Delta_2 - \Delta_3 + d}{2}\right](p_1^2)^{\Delta_1 - d/2}$$
$$\times \left[(p_3^2)^{(\Delta_3 - \Delta_1 - \Delta_2)/2}(p_2^2)^{\Delta_2 - d/2}G_{\Delta_1\Delta_2,\Delta_3}\left(\frac{p_1^2}{p_3^2}, \frac{p_2^2}{p_3^2}\right)\right.$$
$$\left. + (p_3^2)^{(\Delta_3 - \Delta_1 + \Delta_2 - d)/2}G_{\Delta_1\widetilde{\Delta}_2,\Delta_3}\left(\frac{p_1^2}{p_3^2}, \frac{p_2^2}{p_3^2}\right)\right].$$
(4.23)

All other real configurations lead to vanishing Wightman functions by the spectral condition. Note that the correlators (4.21) and (4.23) are related by Hermitian conjugation, since for real operators $\phi(p)^\dagger = \phi(-p)$. Once again we observe that the coefficient $C$ must be real, and as a consequence the Wightman function are real too.

In fact, $C$ can be determined by comparison with the results of a direct Fourier transform performed in ref. [11]: if we adopt the standard convention that the Wightman 3-point function in position space is

$$\langle 0|\,\widetilde{\phi}(x_1)\widetilde{\phi}(x_2)\widetilde{\phi}(x_3)\,|0\rangle = \frac{\lambda}{(x_{12}^2)^{(\Delta_1 + \Delta_2 - \Delta_3)/2}(x_{13}^2)^{(\Delta_1 + \Delta_3 - \Delta_2)/2}(x_{23}^2)^{(\Delta_2 + \Delta_3 - \Delta_1)/2}},$$
(4.24)

where $\lambda$ is the real OPE coefficient and we have denoted $x_{ab}^2 = -(x_a^0 - x_b^0 + i\varepsilon)^2 + (\mathbf{x}_a - \mathbf{x}_b)^2$, then we have

$$C = -\frac{\lambda (4\pi)^d}{2^{\Delta_1 + \Delta_2 + \Delta_3} \Gamma\left(\frac{\Delta_1 + \Delta_2 - \Delta_3}{2}\right) \Gamma\left(\frac{\Delta_1 + \Delta_3 - \Delta_2}{2}\right) \Gamma\left(\frac{\Delta_2 + \Delta_3 - \Delta_1}{2}\right) \Gamma\left(\frac{\Delta_1 + \Delta_2 + \Delta_3 - d}{2}\right)}. \quad (4.25)$$

This coefficient is symmetric under the exchange of the scaling dimensions $\Delta_i$, in agreement with what we found before. Some of the $\Gamma$-functions in the denominator cancel with some of those included in the definition (3.21) of the function $G$. Remarkably, the remaining $\Gamma$-functions in $G$ that have poles at $\Delta_{1,3} = \frac{d}{2} + n$ with integer $n$ also combine with the sines present in the results (4.20) to (4.23) to form a result that is analytic in all $\Delta_i$. For instance, the correlator (4.20) equates

$$\langle\!\langle \phi_1(-p_1)\phi_2(q_2)\phi_3(p_3) \rangle\!\rangle$$
$$= \frac{\lambda (4\pi)^{d+2}(-q_2^2)^{(\Delta_2 - \Delta_1 - \Delta_3)/2}(p_1^2)^{\Delta_1 - d/2}(p_3^2)^{\Delta_3 - d/2}}{2^{\Delta_1 + \Delta_2 + \Delta_3 + 2}\Gamma\left(\Delta_1 - \frac{d}{2} + 1\right)\Gamma\left(\Delta_3 - \frac{d}{2} + 1\right)\Gamma\left(\frac{\Delta_1 + \Delta_2 - \Delta_3}{2}\right)\Gamma\left(\frac{\Delta_2 + \Delta_3 - \Delta_1}{2}\right)}$$
$$\times F_4\left(\frac{\Delta_1 + \Delta_3 - \Delta_2}{2}, \frac{\Delta_1 + \Delta_2 + \Delta_3 - d}{2}, \Delta_1 - \frac{d}{2} + 1, \Delta_3 - \frac{d}{2} + 1; \frac{p_1^2}{q_2^2}, \frac{p_3^2}{q_2^2}\right), \quad (4.26)$$

which is analytic in all the $\Delta_i$ satisfying the unitarity bound ($\Delta_i \geq \frac{d}{2} - 1$). The correlators (4.21) and (4.23) are similarly analytic in all $\Delta_i$, even though their analyticity in $\Delta_2$ is less obvious as it involves relations between the two Appell $F_4$ functions. Note also that some simplifications occur in the Wightman function when the scaling dimensions obey relations of the form $\Delta_a = \Delta_b + \Delta_c + 2n$ with integer $n$. These simplifications are consistent with the factorization occurring in generalized free field theory [11].

# 5 T-products and the Schwinger function

Besides Wightman functions, time-ordered correlators form another important class of observables in quantum field theory, as they are generated by the Lorentzian path integral. In this section we construct time-ordered products (or T-products) purely in terms of R-products and of Wightman function. We also show how they are related by a Wick rotation to the Euclidean Schwinger function.

## 5.1 Partial time-ordering

Let us begin with the T-product of two operators. By analogy with eq. (2.27), one can show that

$$\langle\!\langle T[\phi_1(k_1)\phi_2(k_2)]\phi_3(p_3) \rangle\!\rangle = \langle\!\langle R[\phi_1(k_1), \phi_2(k_2)]\phi_3(p_3) \rangle\!\rangle + \langle\!\langle \phi_2(k_2)\phi_1(k_1)\phi_3(p_3) \rangle\!\rangle$$
$$= \langle\!\langle R[\phi_2(k_2), \phi_1(k_1)]\phi_3(p_3) \rangle\!\rangle + \langle\!\langle \phi_1(k_1)\phi_2(k_2)\phi_3(p_3) \rangle\!\rangle. \quad (5.1)$$

Using either of the two equalities, one can see that the 2-point T-product is in most cases equivalent to an R-product. This is the case when both $k_1$ and $k_2$ are space-like,

$$\langle\!\langle T[\phi_1(q_1)\phi_2(q_2)]\phi_3(p_3) \rangle\!\rangle = \langle\!\langle R[\phi_1(q_1), \phi_2(q_2)]\phi_3(p_3) \rangle\!\rangle \quad \rightarrow \quad \text{eq. (4.4)}, \quad (5.2)$$

when $k_1$ or $k_2$ is time-like,

$$\langle\!\langle \mathrm{T}[\phi_1(-p_1)\phi_2(q_2)]\phi_3(p_3)\rangle\!\rangle = \langle\!\langle \mathrm{R}[\phi_1(-p_1),\phi_2(q_2)]\phi_3(p_3)\rangle\!\rangle \quad\rightarrow\quad \text{eq. (4.9)}, \qquad (5.3)$$

and even when $k_1$ and $k_2$ are both time-like, provided that they point in opposite directions,

$$\langle\!\langle \mathrm{T}[\phi_1(-p_1)\phi_2(p_2)]\phi_3(p_3)\rangle\!\rangle = \langle\!\langle \mathrm{R}[\phi_1(-p_1),\phi_2(p_2)]\phi_3(p_3)\rangle\!\rangle \quad\rightarrow\quad \text{eq. (4.12)}. \qquad (5.4)$$

Note that the T-product is symmetric so the ordering of the operators $\phi_1$ and $\phi_2$ is irrelevant on the left-hand side of these equations, but it is not on the right-hand side. In the case where $k_1$ and $k_2$ are both time-like and backward-directed, one needs to use both the retarded commutator (4.13) and the Wightman function (4.23) to get

$$\langle\!\langle \mathrm{T}[\phi_1(-p_1)\phi_2(-p_2)]\phi_3(p_3)\rangle\!\rangle$$
$$= 2iC\Bigg[(p_3^2)^{(\Delta_3-\Delta_1-\Delta_2)/2}(p_1^2)^{\Delta_1-d/2}(p_2^2)^{\Delta_2-d/2}G_{\Delta_1\Delta_2,\Delta_3}\left(\frac{p_1^2}{p_3^2},\frac{p_2^2}{p_3^2}\right)$$
$$\times i\left(e^{i\pi(\Delta_1+\Delta_2-\Delta_3)/2}\cos\left[\pi(\Delta_1-\Delta_2)\right]-\cos\left[\pi\tfrac{\Delta_1+\Delta_2+\Delta_3-2d}{2}\right]\right)$$
$$+ (p_3^2)^{(\Delta_3-\Delta_1+\Delta_2-d)/2}(p_1^2)^{\Delta_1-d/2}G_{\Delta_1\widetilde{\Delta}_2,\Delta_3}\left(\frac{p_1^2}{p_3^2},\frac{p_2^2}{p_3^2}\right)\sin\left(\pi\tfrac{\Delta_3-\Delta_1+\Delta_2-d}{2}\right)e^{i\pi(\Delta_1-d/2)}$$
$$+ (p_3^2)^{(\Delta_3+\Delta_1-\Delta_2-d)/2}(p_2^2)^{\Delta_2-d/2}G_{\widetilde{\Delta}_1\Delta_2,\Delta_3}\left(\frac{p_1^2}{p_3^2},\frac{p_2^2}{p_3^2}\right)\sin\left(\pi\tfrac{\Delta_3+\Delta_1-\Delta_2-d}{2}\right)e^{i\pi(\Delta_2-d/2)}$$
$$+ (p_3^2)^{(\Delta_3+\Delta_1+\Delta_2-2d)/2}G_{\widetilde{\Delta}_1\widetilde{\Delta}_2,\Delta_3}\left(\frac{p_1^2}{p_3^2},\frac{p_2^2}{p_3^2}\right)\sin\left(\pi\tfrac{\Delta_3+\Delta_1+\Delta_2-2d}{2}\right)\Bigg]. \qquad (5.5)$$

Similar results can be obtained when the time-ordered product is to the right of the third operator, but they can also be related to our previous findings in the following way: since by definition, using the identity (2.26),

$$\langle\!\langle \phi_1(-p_1)\,\mathrm{T}[\phi_2(k_2)\phi_3(k_3)]\rangle\!\rangle = \langle\!\langle \phi_1(-p_1)\,\mathrm{R}[\phi_2(k_2),\phi_3(k_3)]\rangle\!\rangle + \langle\!\langle \phi_1(-p_1)\phi_3(k_3)\phi_2(k_2)\rangle\!\rangle, \qquad (5.6)$$

one can use eq. (4.18) and the reality of the Wightman 3-point function to write

$$\langle\!\langle \phi_1(-p_1)\,\mathrm{T}[\phi_2(k_2)\phi_3(k_3)]\rangle\!\rangle = \langle\!\langle \mathrm{R}[\phi_3(-k_3),\phi_2(-k_2)]\phi_1(p_1)\rangle\!\rangle + \langle\!\langle \phi_2(-k_2)\phi_3(-k_3)\phi_1(p_1)\rangle\!\rangle, \qquad (5.7)$$

which implies

$$\langle\!\langle \phi_1(-p_1)\,\mathrm{T}[\phi_2(k_2)\phi_3(k_3)]\rangle\!\rangle = \langle\!\langle \mathrm{T}[\phi_3(-k_3)\phi_2(-k_2)]\phi_1(p_1)\rangle\!\rangle. \qquad (5.8)$$

These correlators complement the results of ref. [11] in which the partially-time-ordered 3-point function has been calculated in some cases but not in all generality.

Note that one can also define an anti-time-ordered product (denoted $\overline{\mathrm{T}}$ below) in which the operators are ordered in the opposite way as in the time-ordered product. It is defined in terms of retarded commutator by

$$\langle\!\langle \overline{\mathrm{T}}[\phi_1(k_1)\phi_2(k_2)]\phi_3(p_3)\rangle\!\rangle = -\langle\!\langle \mathrm{R}[\phi_1(k_1),\phi_2(k_2)]\phi_3(p_3)\rangle\!\rangle + \langle\!\langle \phi_1(k_1)\phi_2(k_2)\phi_3(p_3)\rangle\!\rangle. \qquad (5.9)$$

Using the identity (4.16) and the reality of the Wightman function, this implies that the anti-time-ordered product is the complex conjugate of the time-ordered product,

$$\langle\!\langle \overline{\mathrm{T}}[\phi_1(k_1)\phi_2(k_2)]\phi_3(p_3)\rangle\!\rangle = \langle\!\langle \mathrm{T}[\phi_1(k_1)\phi_2(k_2)]\phi_3(p_3)\rangle\!\rangle^*. \qquad (5.10)$$

Conformal correlation functions involving pair-wise (anti-)time-ordered operators were used in refs. [25, 26] to obtain sum rules for anomaly coefficients.

## 5.2   The 3-point T-product

Finally, the last type of correlation function in Minkowski momentum space is the time-ordered product of 3 operators. In position space, the T-product is defined as

$$\langle 0 | \, T[\widetilde{\phi}_1(x_1)\widetilde{\phi}(x_2)\widetilde{\phi}_3(x_3)] \, |0\rangle = \sum_\sigma \theta(x^0_{\sigma(1)} - x^0_{\sigma(2)})\theta(x^0_{\sigma(2)} - x^0_{\sigma(3)})$$

$$\times \langle 0 | \, \widetilde{\phi}_{\sigma(1)}(x_{\sigma(1)})\widetilde{\phi}_{\sigma(2)}(x_{\sigma(2)})\widetilde{\phi}_{\sigma(3)}(x_{\sigma(3)}) \, |0\rangle$$

$$+ \text{ contact terms}, \tag{5.11}$$

where the sum is over all permutations of the labels $\{1, 2, 3\}$. Using $\theta(a) + \theta(-a) = 1$ together with

$$\theta(a - b)\theta(b - c) + \theta(a - c)\theta(c - b) + \theta(c - a)\theta(a - b) = \theta(a - b), \tag{5.12}$$

one can prove the following relation

$$\langle 0 | \, T[\widetilde{\phi}_1(x_1)\widetilde{\phi}_2(x_2)\widetilde{\phi}_3(x_3)] \, |0\rangle$$

$$= \langle 0 | \, R[\widetilde{\phi}_1(x_1), \widetilde{\phi}_2(x_2)\widetilde{\phi}_3(x_3)] \, |0\rangle + \langle 0 | \, T[\widetilde{\phi}_2(x_2), \widetilde{\phi}_3(x_3)] \, \widetilde{\phi}_1(x_1) \, |0\rangle$$

$$+ \langle 0 | \, \widetilde{\phi}_2(x_2) \, R[\widetilde{\phi}_1(x_1), \widetilde{\phi}_3(x_3)] \, |0\rangle + \langle 0 | \, \widetilde{\phi}_3(x_3) \, R[\widetilde{\phi}_1(x_1), \widetilde{\phi}_2(x_2)] \, |0\rangle . \tag{5.13}$$

This is a generalization to 3 points of the identity (2.26). Once again, there are no contact terms in this expression as both sides of the equality are already tempered distributions. When Fourier transformed, it defines the time-ordered 3-point function in terms of correlators that we have already determined,

$$\langle\!\langle T[\phi_1(k_1)\phi_2(k_2)\phi_3(k_3)]\rangle\!\rangle = \langle\!\langle R[\phi_1(k_1), \phi_2(k_2)\phi_3(k_3)]\rangle\!\rangle + \langle\!\langle T[\phi_2(k_2), \phi_3(k_3)]\phi_1(k_1)\rangle\!\rangle$$

$$+ \langle\!\langle \phi_2(k_2) \, R[\phi_1(k_1), \phi_3(k_3)]\rangle\!\rangle + \langle\!\langle \phi_3(k_3) \, R[\phi_1(k_1), \phi_2(k_2)]\rangle\!\rangle . \tag{5.14}$$

When all 3 momenta are space-like, the T-product coincides therefore with the R-product,

$$\langle\!\langle T[\phi_1(q_1)\phi_2(q_2)\phi_3(q_3)]\rangle\!\rangle = \langle\!\langle R[\phi_1(q_1), \phi_2(q_2)\phi_3(q_3)]\rangle\!\rangle \quad \rightarrow \quad \text{eq. (3.38).} \tag{5.15}$$

Similarly, when only one momentum is time-like, we can use

$$\langle\!\langle T[\phi_1(q_1)\phi_2(q_2)\phi_3(p_3)]\rangle\!\rangle = \langle\!\langle R[\phi_1(q_1), \phi_2(q_2)\phi_3(p_3)]\rangle\!\rangle \quad \rightarrow \quad \text{eq. (3.35)} \tag{5.16}$$

and

$$\langle\!\langle T[\phi_1(q_1)\phi_2(q_2)\phi_3(-p_3)]\rangle\!\rangle = \langle\!\langle R[\phi_3(-p_3), \phi_1(q_1)\phi_2(p_2)]\rangle\!\rangle \quad \rightarrow \quad \text{eq. (3.36).} \tag{5.17}$$

Remarkably, the right-hand sides of eqs. (5.16) and (5.17) are equal: independently of whether the time-like momentum is forward- or backward-directed, we have

$$\langle\!\langle T[\phi_1(q_1)\phi_2(q_2)\phi_3(\pm p_3)]\rangle\!\rangle$$

$$= C\Bigg[ (p_3^2)^{(\Delta_3 - \Delta_1 - \Delta_2)/2}(-q_1^2)^{\Delta_1 - d/2}(-q_2^2)^{\Delta_2 - d/2} G_{\Delta_1 \Delta_2, \Delta_3}\left(\frac{q_1^2}{p_3^2}, \frac{q_2^2}{p_3^2}\right) e^{i\pi(\Delta_3 - \Delta_1 - \Delta_2)/2}$$

$$+ (p_3^2)^{(\Delta_3 - \Delta_1 + \Delta_2 - d)/2}(-q_1^2)^{\Delta_1 - d/2} G_{\Delta_1 \widetilde{\Delta}_2, \Delta_3}\left(\frac{q_1^2}{p_3^2}, \frac{q_2^2}{p_3^2}\right) e^{i\pi(\Delta_3 - \Delta_1 + \Delta_2 - d)/2}$$

$$+ (p_3^2)^{(\Delta_3 + \Delta_1 - \Delta_2 - d)/2}(-q_2^2)^{\Delta_2 - d/2} G_{\widetilde{\Delta}_1 \Delta_2, \Delta_3}\left(\frac{q_1^2}{p_3^2}, \frac{q_2^2}{p_3^2}\right) e^{i\pi(\Delta_3 + \Delta_1 - \Delta_2 - d)/2}$$

$$+ (p_3^2)^{(\Delta_3 + \Delta_1 + \Delta_2 - 2d)/2} G_{\widetilde{\Delta}_1 \widetilde{\Delta}_2, \Delta_3}\left(\frac{q_1^2}{p_3^2}, \frac{q_2^2}{p_3^2}\right) e^{i\pi(\Delta_3 + \Delta_1 + \Delta_2 - 2d)/2} \Bigg] . \tag{5.18}$$

When two momenta are time-like, we find

$$
\begin{aligned}
\langle\!\langle \mathrm{T}[\phi_1(q_1)\phi_2(-p_2)\phi_3(p_3)]\rangle\!\rangle &= \langle\!\langle \mathrm{R}[\phi_2(-p_2),\phi_1(q_1)\phi_3(p_3)]\rangle\!\rangle \\
&= C\bigg[ (-q_1^2)^{(\Delta_1-\Delta_2-\Delta_3)/2}(p_2^2)^{\Delta_2-d/2}(p_3^2)^{\Delta_3-d/2}G_{\Delta_2\Delta_3,\Delta_1}\left(\frac{p_2^2}{q_1^2},\frac{p_3^2}{q_1^2}\right)e^{i\pi(\Delta_2+\Delta_3-d)/2} \\
&\quad + (-q_1^2)^{(\Delta_1-\Delta_2+\Delta_3-d)/2}(p_2^2)^{\Delta_2-d/2}G_{\Delta_2\widetilde{\Delta}_3,\Delta_1}\left(\frac{p_2^2}{q_1^2},\frac{p_3^2}{q_1^2}\right)e^{i\pi(\Delta_2-d/2)/2} \\
&\quad + (-q_1^2)^{(\Delta_1+\Delta_2-\Delta_3-d)/2}(p_3^2)^{\Delta_3-d/2}G_{\widetilde{\Delta}_2\Delta_3,\Delta_1}\left(\frac{p_2^2}{q_1^2},\frac{p_3^2}{q_1^2}\right)e^{i\pi(\Delta_3-d/2)/2} \\
&\quad + (-q_1^2)^{(\Delta_1+\Delta_2+\Delta_3-2d)/2}G_{\widetilde{\Delta}_2\widetilde{\Delta}_3,\Delta_1}\left(\frac{p_2^2}{q_1^2},\frac{p_3^2}{q_1^2}\right)\bigg].
\end{aligned}
\tag{5.19}
$$

Finally, when all three momenta are time-like we can use

$$
\langle\!\langle \mathrm{T}[\phi_1(-p_1)\phi_2(p_2)\phi_3(p_3)]\rangle\!\rangle = \langle\!\langle \mathrm{R}[\phi_1(-p_1),\phi_2(p_2)\phi_3(p_3)]\rangle\!\rangle \quad\rightarrow\quad \text{eq. (3.42).} \tag{5.20}
$$

If instead two of the momenta are backward-directed, then we need to take a linear combination of a R-product with a Wightman function to compute the T-product, but the result turns out to be identical,

$$
\begin{aligned}
\langle\!\langle \mathrm{T}[\phi_1(-p_1)\phi_2(\pm p_2)\phi_3(p_3)]\rangle\!\rangle &= C\, e^{i\pi(\Delta_1+\Delta_2+\Delta_3-2d)/2} \\
&\times\bigg[ (p_1^2)^{(\Delta_1-\Delta_2-\Delta_3)/2}(p_2^2)^{\Delta_2-d/2}(p_3^2)^{\Delta_3-d/2}G_{\Delta_2\Delta_3,\Delta_1}\left(\frac{p_2^2}{p_1^2},\frac{p_3^2}{p_1^2}\right) \\
&\quad + (p_1^2)^{(\Delta_1-\Delta_2+\Delta_3-d)/2}(p_2^2)^{\Delta_2-d/2}G_{\Delta_2\widetilde{\Delta}_3,\Delta_1}\left(\frac{p_2^2}{p_1^2},\frac{p_3^2}{p_1^2}\right) \\
&\quad + (p_1^2)^{(\Delta_1+\Delta_2-\Delta_3-d)/2}(p_3^2)^{\Delta_3-d/2}G_{\widetilde{\Delta}_2\Delta_3,\Delta_1}\left(\frac{p_2^2}{p_1^2},\frac{p_3^2}{p_1^2}\right) \\
&\quad + (p_1^2)^{(\Delta_1+\Delta_2+\Delta_3-2d)/2}G_{\widetilde{\Delta}_2\widetilde{\Delta}_3,\Delta_1}\left(\frac{p_2^2}{p_1^2},\frac{p_3^2}{p_1^2}\right)\bigg].
\end{aligned}
\tag{5.21}
$$

Note that all four cases (5.15), (5.18), (5.19) and (5.21) can be compactly summarized into one expression,

$$
\begin{aligned}
\langle\!\langle &\mathrm{T}[\phi_1(k_1)\phi_2(k_2)\phi_3(k_3)]\rangle\!\rangle \\
&= C\bigg[ [-k_3^2]_F^{(\Delta_3-\Delta_1-\Delta_2)/2}[-k_1^2]_F^{\Delta_1-d/2}[-k_2^2]_F^{\Delta_2-d/2}G_{\Delta_1\Delta_2,\Delta_3}\left(\frac{[-k_1^2]_F}{[-k_3^2]_F},\frac{[-k_2^2]_F}{[-k_3^2]_F}\right) \\
&\quad + [-k_3^2]_F^{(\Delta_3+\Delta_1-\Delta_1-d)/2}[-k_1^2]_F^{\Delta_1-d/2}G_{\Delta_1\widetilde{\Delta}_2,\Delta_3}\left(\frac{[-k_1^2]_F}{[-k_3^2]_F},\frac{[-k_2^2]_F}{[-k_3^2]_F}\right) \\
&\quad + [-k_3^2]_F^{(\Delta_3+\Delta_1-\Delta_2-d)/2}[-k_2^2]_F^{\Delta_2-d/2}G_{\widetilde{\Delta}_1\Delta_2,\Delta_3}\left(\frac{[-k_1^2]_F}{[-k_3^2]_F},\frac{[-k_2^2]_F}{[-k_3^2]_F}\right) \\
&\quad + [-k_3^2]_F^{(\Delta_3+\Delta_1+\Delta_2-2d)/2}G_{\widetilde{\Delta}_1\widetilde{\Delta}_2,\Delta_3}\left(\frac{[-k_1^2]_F}{[-k_3^2]_F},\frac{[-k_2^2]_F}{[-k_3^2]_F}\right)\bigg],
\end{aligned}
\tag{5.22}
$$

where we have introduced the "Feynman $i\varepsilon$" prescription,

$$
[-k_i^2]_F^\alpha = \lim_{\varepsilon\to 0_+}(-k_i^2+i\varepsilon)^\alpha = \begin{cases} (-k_i^2)^\alpha & \text{for } k_i^2 \le 0, \\ e^{i\pi\alpha}(k_i^2)^\alpha & \text{for } k_i^2 > 0. \end{cases} \tag{5.23}
$$

This correlation function involves the same linear combination of Appell $F_4$ functions as the R-product at space-like momenta, eq. (3.38), and it is therefore also regular for all values of their arguments. The only singularities are light-cone ones, whenever $k_i^2 = 0$ for some $i = 1, 2, 3$, and the $i\varepsilon$ prescription tells us precisely how to deal with them. For instance, this illustrates how the conformal LSZ reduction defined in ref. [27] works at the level of 3-point functions.

## 5.3 Wick rotation to Euclidean space

The result eq. (5.22) is in fact reminiscent of perturbative scattering amplitude computations in which squares of the momenta always appear with the Feynman $i\varepsilon$ prescription. In this case, it is standard practice to perform Euclidean integrals after taking the Minkowskian expression through a Wick rotation.

The same strategy can be applied here. For this, it is most convenient to view the time-ordered correlator as a function of three independent momenta,

$$\langle 0| \, T[\phi_1(k_1)\phi_2(k_2)\phi_3(k_3)] \, |0\rangle = (2\pi)^d \delta^d(k_1 + k_2 + k_3) \langle\!\langle T[\phi_1(k_1)\phi_2(k_2)\phi_3(k_3)]\rangle\!\rangle. \quad (5.24)$$

In terms of the energy component of the momenta, all singularities of eq. (5.22) are found at $k_i^0 = \pm(|\mathbf{k}_i| + i\varepsilon)$, that is in the upper-half complex plane when $k_i^0 > 0$ and in the lower-half complex plane when $k_i^0 < 0$. Therefore, provided that the correlation function decays fast enough as $|k_i^0| \to \infty$, we can perform a Wick rotation that takes $k_i^0 \to -i k_i^d$, simultaneously for all $i = 1, 2, 3$, without encountering any singularity. This continues the time-ordered correlation function to the Euclidean regime, with $-k_i^2 + i\varepsilon \to (k_i^1)^2 + \ldots + (k_i^d)^2$. The rotation of the delta function yields a factor of $i$ that combines with a factor of $(-i)^3$ arising from the mismatch between the definitions of the Minkowskian and Euclidean Fourier transforms, and we arrive finally at

$$\langle \phi_1(k_1)\phi_2(k_2)\phi_3(k_3)\rangle_E = -C \, (2\pi)^d \delta^d(k_1 + k_2 + k_3)$$
$$\times \left[ (k_3^2)^{(\Delta_3 - \Delta_1 - \Delta_2)/2}(k_1^2)^{\Delta_1 - d/2}(k_2^2)^{\Delta_2 - d/2} G_{\Delta_1 \Delta_2, \Delta_3}\left(\frac{k_1^2}{k_3^2}, \frac{k_2^2}{k_3^2}\right) \right.$$
$$+ (k_3^2)^{(\Delta_3 + \Delta_1 - \Delta_1 - d)/2}(k_1^2)^{\Delta_1 - d/2} G_{\Delta_1 \widetilde{\Delta}_2, \Delta_3}\left(\frac{k_1^2}{k_3^2}, \frac{k_2^2}{k_3^2}\right)$$
$$+ (k_3^2)^{(\Delta_3 + \Delta_1 - \Delta_2 - d)/2}(k_2^2)^{\Delta_2 - d/2} G_{\widetilde{\Delta}_1 \Delta_2, \Delta_3}\left(\frac{k_1^2}{k_3^2}, \frac{k_2^2}{k_3^2}\right)$$
$$\left. + (k_3^2)^{(\Delta_3 + \Delta_1 + \Delta_2 - 2d)/2} G_{\widetilde{\Delta}_1 \widetilde{\Delta}_2, \Delta_3}\left(\frac{k_1^2}{k_3^2}, \frac{k_2^2}{k_3^2}\right) \right]. \quad (5.25)$$

This matches precisely the known value of the Euclidean Fourier transform that can be found in refs. [13, 14], including the normalization (4.25) of the OPE coefficient.

# 6 Conclusions

In this work, we have listed exhaustively all conformal 3-point functions of scalar primary operators in Minkowski momentum space, and given in each case a closed-form expression for the correlator in terms of Appell $F_4$ double hypergeometric functions. These results

| | Correlation function | Equation number |
|---|---|---|
| **Wightman** | $\langle\!\langle\phi_1(-p_1)\phi_2(q_2)\phi_3(p_3)\rangle\!\rangle$ | (4.20) |
| | $\langle\!\langle\phi_1(-p_1)\phi_2(p_2)\phi_3(p_3)\rangle\!\rangle$ | (4.21) |
| | $\langle\!\langle\phi_1(-p_1)\phi_2(-p_2)\phi_3(p_3)\rangle\!\rangle$ | (4.23) |
| **T-products** | $\langle\!\langle\mathrm{T}[\phi_1(q_1)\phi_2(q_2)\phi_3(q_3)]\rangle\!\rangle$ | (5.15) $\rightarrow$ (3.38) |
| | $\langle\!\langle\mathrm{T}[\phi_1(q_1)\phi_2(q_2)\phi_3(\pm p_3)]\rangle\!\rangle$ | (5.18) |
| | $\langle\!\langle\mathrm{T}[\phi_1(q_1)\phi_2(-p_2)\phi_3(p_3)]\rangle\!\rangle$ | (5.19) |
| | $\langle\!\langle\mathrm{T}[\phi_1(-p_1)\phi_2(\pm p_2)\phi_3(p_3)]\rangle\!\rangle$ | (5.21) |
| **2-pt T-product** | $\langle\!\langle\mathrm{T}[\phi_1(q_1)\phi_2(q_2)]\phi_3(p_3)\rangle\!\rangle$ | (5.2) $\rightarrow$ (4.4) |
| | $\langle\!\langle\mathrm{T}[\phi_1(-p_1)\phi_2(q_2)]\phi_3(p_3)\rangle\!\rangle$ | (5.3) $\rightarrow$ (4.9) |
| | $\langle\!\langle\mathrm{T}[\phi_1(-p_1)\phi_2(p_2)]\phi_3(p_3)\rangle\!\rangle$ | (5.4) $\rightarrow$ (4.12) |
| | $\langle\!\langle\mathrm{T}[\phi_1(-p_1)\phi_2(-p_2)]\phi_3(p_3)\rangle\!\rangle$ | (5.5) |
| **R-products** | $\langle\!\langle\mathrm{R}[\phi_1(q_1),\phi_2(q_2)\phi_3(q_3)]\rangle\!\rangle$ | (3.38) |
| | $\langle\!\langle\mathrm{R}[\phi_1(q_1),\phi_2(q_2)\phi_3(\pm p_3)]\rangle\!\rangle$ | (3.35) |
| | $\langle\!\langle\mathrm{R}[\phi_1(\pm p_1),\phi_2(q_2)\phi_3(q_3)]\rangle\!\rangle$ | (3.36) |
| | $\langle\!\langle\mathrm{R}[\phi_1(q_1),\phi_2(\pm p_2)\phi_3(\mp p_3)]\rangle\!\rangle$ | (3.40) |
| | $\langle\!\langle\mathrm{R}[\phi_1(\pm p_1),\phi_2(q_2)\phi_3(\mp p_3)]\rangle\!\rangle$ | (3.37) |
| | $\langle\!\langle\mathrm{R}[\phi_1(\pm p_1),\phi_2(\pm p_2)\phi_3(\mp p_3)]\rangle\!\rangle$ | (3.41) |
| | $\langle\!\langle\mathrm{R}[\phi_1(\pm p_1),\phi_2(\mp p_2)\phi_3(\mp p_3)]\rangle\!\rangle$ | (3.42) |
| **Retarded commutator** | $\langle\!\langle\mathrm{R}[\phi_1(q_1),\phi_2(q_2)]\phi_3(p_3)\rangle\!\rangle$ | (4.4) |
| | $\langle\!\langle\mathrm{R}[\phi_1(-p_1),\phi_2(q_2)]\phi_3(p_3)\rangle\!\rangle$ | (4.9) |
| | $\langle\!\langle\mathrm{R}[\phi_1(q_1),\phi_2(-p_2)]\phi_3(p_3)\rangle\!\rangle$ | (4.10) |
| | $\langle\!\langle\mathrm{R}[\phi_1(p_1),\phi_2(-p_2)]\phi_3(p_3)\rangle\!\rangle$ | (4.11) |
| | $\langle\!\langle\mathrm{R}[\phi_1(-p_1),\phi_2(p_2)]\phi_3(p_3)\rangle\!\rangle$ | (4.12) |
| | $\langle\!\langle\mathrm{R}[\phi_1(-p_1),\phi_2(-p_2)]\phi_3(p_3)\rangle\!\rangle$ | (4.13) |

Table 1: List of all non-vanishing Minkowskian correlation functions, with the corresponding equation number. As a reminder, $p_i$ indicates a vector inside the forward light cone and $q_i$ a space-like vector. The function $G_{\Delta_a\Delta_b,\Delta_c}$ appearing in all these equations is defined in eq. (3.21) in terms of the Appell $F_4$ series (3.16). Correlation functions involving a retarded commutator or a 2-point T-product to the right of the third operator can be obtained from this table using the identities (4.18) and (5.8).

can easily be implemented numerically, or studied analytically for instance in the limit in which one or two momenta are light-like. For the reader's convenience, we provide in table 1 a list of all non-vanishing correlators with the corresponding equation number. Among these correlators, the Wightman functions (including the cases with 2-point T- or R-products) are particularly interesting as they form the basis of the OPE in Minkowski momentum space: the function

$$\langle\!\langle \phi_0(-k_1 - k_2)[\phi_1(k_1)\phi_2(k_2)]\rangle\!\rangle, \tag{6.1}$$

where the bracket can mean any particular ordering, computes the overlap of the state $[\phi_1(k_1)\phi_2(k_2)]|0\rangle$, with the state $\phi_0(k_1 + k_2)|0\rangle$ of equal total momentum. For this reason, such correlators are used as building blocks for the conformal partial wave expansion of higher-point functions [28, 29].

Besides these explicit results for scalar 3-point functions, our construction is also an interesting playground for the study of analyticity properties of conformal correlators. The study of micro-causality in CFT has a long history (see e.g. [30]), but the related analyticity properties in momentum space are far from being fully explored. Some results have been obtained resurrecting old QFT tools [31, 32], but to the best of the author's knowledge this is the first time that the analyticity properties of correlation functions are being used with the full power of conformal symmetry, to the point of determining all 2- and 3-point function up to dynamical CFT data. Undoubtedly, the study of analyticity of CFT correlators becomes much more interesting starting with 4-point functions, as it can possibly be used together with the OPE to obtain bounds on the CFT data. We leave this prospect for future studies, but refer the reader to ref. [22] taking interesting steps in this direction. Note also that the ability to go from Minkowskian correlation functions to Euclidean ones is central in the recent cosmological bootstrap attempt to construct inflationary correlators in de Sitter space-time non-perturbatively [33–36].

Finally, we must also recognize that this work on conformal 3-point functions in Minkowski momentum space is far from being complete, as we have made two considerably simplifying assumptions: the primary operators that we consider are all scalars, and their scaling dimension is assumed to stay away from the special cases $\Delta_i = \frac{d}{2} + n$ with integer $n$. In fact, for scalar operators this second issue can be resolved by analytic continuation in the scaling dimensions: we refer the reader to ref. [37] for an in-depth discussion of the problem. Note that the renormalization that is needed in this case only affects the T- and R-products, whereas the Wightman functions are found to be analytic in all scaling dimensions. Regarding spinning operators, very little is known in the general case. In Euclidean momentum-space, various correlation functions involving conserved currents and energy-momentum tensors have been computed [38–41], but no such results exist for generic operators and/or in Minkowski signature. In fact, extending our construction to spinning operator is an interesting open problem. The conformal Ward identities are modified in the presence of spin, and they generically admit multiple solutions, but the axioms that we invoke are left unchanged. In fact, thanks to the ability of projecting momentum eigenstates onto definite polarizations [29], possibly in conjunction with the powerful spinor-helicity formalism [42, 43], the Minkowski momentum-space representation might be particularly well-suited for the study of spinning correlation functions.

## Acknowledgments

The author would like to thank Marco Serone and Slava Rychkov for their helpful comments on the draft.

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
