# Peer review of "From Schwinger to Wightman: all conformal 3-point functions in momentum space"

_SciPost Physics_

## Round 1 · Referee Report · Anonymous (Referee 1) · 2022-12-22

Report

This paper studies the CFT 2- and 3-point functions in momentum space. Understanding of momentum-space CFT correlators is important from various points of view: they encode most directly the spectral conditions, exhibit contact terms, various computations are easier to peform in momentum space.

Since Fourier transform to momentum space is an integral transform, it is very sensitive to the global structure of the position-space correlators, and a separate discussion is needed for different types of correlation functions (Euclidean/Lorentzian T-ordered, Wighman functions). This paper considers all common types by reducing them to various R-products.

R-products in momentum space are determined in this paper from basic symmetry and analyticity/energy-positivty principles rather than from a direct Fourier transform. This not only provides a much more instructive derivation than what would be achieved by a direct Fourier integral, but also gives a useful survey of various properties of correlation functions in momentum space.

The main result of the paper is a comprehensive list of scalar correlation functions in momemtum space. It will certainly serve as an extremely useful reference for the experts in the field. More general correlation functions can be obtained from the results of this paper using standard methods such as weight-shifting operators.

The paper is well-organized and presents the material in a very clear way. One small comment is that the discussion of the role of contact terms could have been somewhat more careful and explicit, but these are, in the end, not the main focus of the paper.

I belive that the paper makes an important technical contribution to the literature and should be published in its present form.

---

## Round 1 · Referee Report · Slava Rychkov (Referee 2) · 2022-12-25

Strengths

  • constructs retarded and time-ordered correlation 2 and 3-point functions in conformal field theories in d dimensions

Weaknesses

  • assumptions and definitions are not clearly stated

Report

This paper gives a construction of conformal 2-point and 3-point time-ordered and retarded functions in conformal field theory.

I found the logic of this paper hard to follow because the assumptions are not carefully stated. The first sentence of the abstract refers to an "axiomatic" way of constructing the correlators, but in fact the paper is more intuitive than axiomatic.

My problems start on p.2 in footnote 1 where the author states that all his assumptions can be traced to Euclidean bootstrap axioms used in [9,10], but that this is unnecessary. That would be fine, but then the author would need to state which assumptions is he actually using. Note that [9,10] left out retarded and time-ordered correlators from considerations. The list (1)-(4) that follows is really too brief from my point of view to serve as a basis, and would need to be expanded.

On p.1 the author states that his assumptions are "essentially equivalent to Wightman axioms, plus the constraints of conformal invariance," only to admit in note 2 that Wightman axioms do not guarantee the existence of R-products as tempered distributions. Why not be more transparent about this important point?

A confusing, for me, point is that the author treats Eqs (2.5) and (3.4) as definitions of R-product. In the axiomatic point of view to which the author professes to adhere, and which I think is quite appropriate in this context, these equations are certainly not definitions, but requirements that R-products need to satisfy. (Note that Ref. [17] is a non-rigorous reference, written before the subtleties concerning the potential non-existence of R-products as tempered distributions was realized. Indeed it predates Wightman axioms. An additional more modern reference would be desired.) Axiomatically, the R-products and T-product correlators are defined as collections of tempered distributions satisfying certain properties, which can be motivated by formal considerations, but eventually have to be postulated, see e.g. the book by Bogolyubov, Logunov and Todorov, Chapter 13 "Lehmann-Symanzik-Zimmermann Formalism"

In section 2.1, the author works out the retarded 2-point function in the forward tube and in the real domain. His result is (2.12) for timelike separation and (2.13) for spacelike separation. What about light-like separation? The retarded 2-point function and its Fourier transform must be tempered distributions. Does it follow from the author's result that they are tempered distribution everywhere, including the lightcone? If not the claim of fully constructing these correlators needs to be moderated. (I do think this can be done using Vladimirov's theorem and powerlaw bound, see [9,10])

Note 4: Which assumption does this follows from?

Note 5. Can one provide a reference?

Eq. (2.22): Can one provide a reference?

Last paragraph of section 2.2: Might I guess what the author intends to say in this paragraph: that for these special scaling dimensions, the R-product may be defined satisfying some but not all requirements that he mentioned beforehand (e.g. one has to abandon conformal invariance). This discussion would benefit from stating the assumptions and the goals more clearly.

In my opinion, the paper would greatly benefit from a cosmetic rewrite which would: - state early on, and in equations not in words, all the requirements on conformal field theory that the author will be using - state early on the axiomatic requirements that R- and T-products need to satisfy. (avoiding the use of the word "definition", since it's far from obvious, as note 2 correctly states, that R- and T-products satisfying these requirements do exist as tempered distributions) - state early on the assumptions of conformal invariance saying that in some cases they will have to be abandoned.

The main result of this paper would then be that R- and T-products satisfying these requirements do actually exist, up and including the 3-point function case. (provided that the author knows how to address what happens on the lightcone, otherwise the claims need to be moderated)

Requested changes

Please see the report

---

## Round 1 · Referee Report · Anonymous (Referee 3) · 2023-1-7

Report

This paper aims to derive conformal 3-point functions of scalar primary operators in momentum space. The construction is based on the solution of conformal Ward identities for retarded products and then use of analytic continuation to obtain 3-point functions for Wightman functions, time-products and Euclidean correlators. The results obtained are for cases that do not require renormalisation.

The authors states properties (1)-(4) as his starting point, but it seems to me the results follow by weaker assumptions: one only needs to require that the Euclidean correlators are tempered distributions and satisfy the conformal Ward identities (and of course that a CFT with the required spectrum exists). The conformal Ward identities for scalar operators have been solved in momentum space in complete generality in Euclidean signature in Ref. [13],[14],[37] (including also the cases that require renormalization). The solution is unique, and it is given in terms of functions with known analytic properties, as discussed in [14], [37] and

A.Bzowski, P. McFadden and K. Skenderis,
``Evaluation of conformal integrals,''
JHEP 02 (2016), 068
doi:10.1007/JHEP02(2016)068
[arXiv:1511.02357 [hep-th]].

It follows that results can be analytically continued to appropriate domains in Minkowski space. Working out the details of this analytic continuation is a non-trivial technical exercise, and one may view this paper as doing this: one may read the paper from the last section to the first, starting from the Euclidean correlator and analytically continuing to Minkowski correlators. In this sense the paper contains useful and publishable results.

Requested changes

1- The author should amend the narrative to reflect the comments in the report and discuss the perspective discussed there and the advantages/disadvantages relatively to using properties (1)-(4) (or adopt this perspective).

2- The author should include in the analysis the cases that require renormalisation. UV infinities are local or semi-local and as such their analytic structure is simple in momentum space, so it would be straightforward to analytically continue them to different domains. [Note that the special cases are not only operators of dimension Delta=d/2 +n, but more generally 3-point functions involving operators with dimension Delta_i such that they satisfy d/2 + Sum_i sigma_i (Delta_i – d/2)= - 2 n, where n is an integer and the signs sigma_i are either all minus or two minus and one plus]. While computationally it would be straightforward to extend the analysis to these cases, the physics implications would most likely be non-trivial and it would be interesting to understand which Lorentzian axioms may need to be relaxed to accommodate, for example, the scale violating products of logarithms that renormalized correlators contain due to anomalies and beta functions (see [37]).

---

## Editorial Decision

awaiting_resubmission